# PIX2SEQ: A LANGUAGE MODELING FRAMEWORK FOR OBJECT DETECTION

**Ting Chen, Saurabh Saxena, Lala Li, David J. Fleet, Geoffrey Hinton**
Google Research, Brain Team

## ABSTRACT

We present *Pix2Seq*, a simple and generic framework for object detection. Unlike existing approaches that explicitly integrate prior knowledge about the task, we cast object detection as a language modeling task conditioned on the observed pixel inputs. Object descriptions (e.g., bounding boxes and class labels) are expressed as sequences of discrete tokens, and we train a neural network to perceive the image and generate the desired sequence. Our approach is based mainly on the intuition that if a neural network knows about where and what the objects are, we just need to teach it how to read them out. Beyond the use of task-specific data augmentations, our approach makes minimal assumptions about the task, yet it achieves competitive results on the challenging COCO dataset, compared to highly specialized and well optimized detection algorithms.[1]

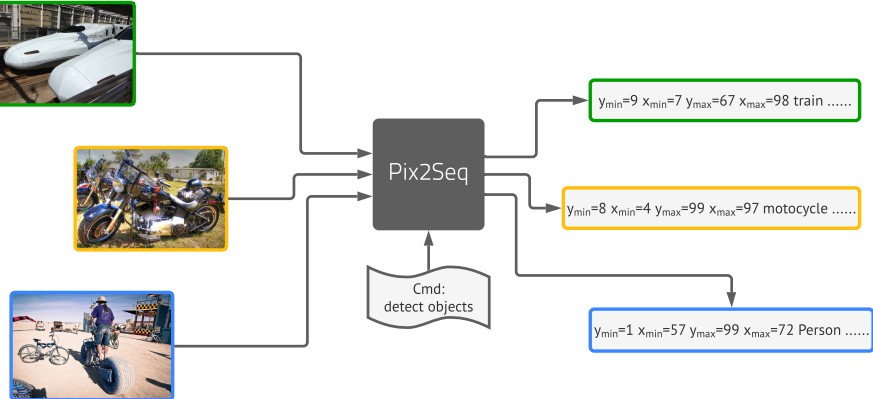

Figure 1: Illustration of Pix2Seq framework for object detection. The neural net perceives an image and generates a sequence of tokens that correspond to bounding boxes and class labels.

## 1 INTRODUCTION

Visual object detection systems aim to recognize and localize all objects of pre-defined categories in an image. The detected objects are typically described by a set of bounding boxes and associated class labels. Given the difficulty of the task, most existing methods, such as (Girshick, 2015; Ren et al., 2015; He et al., 2017; Lin et al., 2017b; Carion et al., 2020), are carefully designed and highly customized, with a significant amount of prior knowledge in the choice of architecture and loss function. For example, many architectures are tailored to the use of bounding boxes (e.g., with region proposals (Girshick, 2015; Ren et al., 2015) and RoI pooling (Girshick et al., 2014; He et al., 2017)). Others are tied to the use of object queries for object binding (Carion et al., 2020). Loss functions are often similarly tailored to the use of bounding boxes, such as box regression (Szegedy et al., 2013; Lin et al., 2017b), set-based matching (Erhan et al., 2014; Carion et al., 2020), or by incorporating

---

Correspondence to: `iamtingchen@google.com`

[1]Code and checkpoints available at https://github.com/google-research/pix2seq.

specific performance metrics, like intersection-over-union on bounding boxes (Rezatofighi et al., 2019). Although existing systems find applications in myriad domains, from self-driving cars (Sun et al., 2020), to medical image analysis (Jaeger et al., 2020), to agriculture (Sa et al., 2016), the specialization and complexity make them difficult to integrate into a larger system, or generalize to a much broader array of tasks associated with general intelligence.

This paper advocates a new approach, based on the intuition that if a neural net knows about where and what the objects are, we just need to teach it to read them out. And by learning to "describe" objects the model can learn to ground the "language" on pixel observations, leading to useful object representations. This is realized with our Pix2Seq framework (see Figure 1). Given an image, our model produces a sequence of discrete tokens that correspond to object descriptions (e.g., object bounding boxes and class labels), reminiscent of an image captioning system (Vinyals et al., 2015b; Karpathy & Fei-Fei, 2015; Xu et al., 2015). In essence, we cast object detection as a language modeling task conditioned on pixel inputs, for which the model architecture and loss function are generic and relatively simple, without being engineered specifically for the detection task. As such, one can readily extend the framework to different domains or applications, or incorporate it into a perceptual system supporting general intelligence, for which it provides a language interface to a wide range of vision tasks.

To tackle the detection task with Pix2Seq, we first propose a quantization and serialization scheme that converts bounding boxes and class labels into sequences of discrete tokens. We then leverage an encoder-decoder architecture for perceiving pixel inputs and generating the target sequence. The objective function is simply the maximum likelihood of tokens conditioned on pixel inputs and the preceding tokens. While both the architecture and loss function are task-agnostic (without assuming prior knowledge about object detection, e.g., bounding boxes), we can still incorporate task-specific prior knowledge with a sequence augmentation technique, proposed below, that alters both input and target sequences during training. Through extensive experimentation, we demonstrate that this simple Pix2Seq framework can achieve competitive results on the COCO dataset compared to highly customized, well established approaches, including Faster R-CNN (Ren et al., 2015) and DETR (Carion et al., 2020). By pretraining our model on a larger object detection dataset, its performance can be further improved.

## 2 THE PIX2SEQ FRAMEWORK

In the proposed Pix2Seq framework we cast object detection as a language modeling task, conditioned on pixel inputs (Figure 1). The system consists of four main components (Figure 2):

- *Image Augmentation*: As is common in training computer vision models, we use image augmentations to enrich a fixed set of training examples (e.g., with random scaling and crops).

- *Sequence construction & augmentation*: As object annotations for an image are usually represented as a *set* of bounding boxes and class labels, we convert them into a *sequence* of discrete tokens.

- *Architecture*: We use an encoder-decoder model, where the encoder perceives pixel inputs, and the decoder generates the target sequence (one token at a time).

- *Objective/loss function*: The model is trained to maximize the log likelihood of tokens conditioned on the image and the preceding tokens (with a softmax cross-entropy loss).

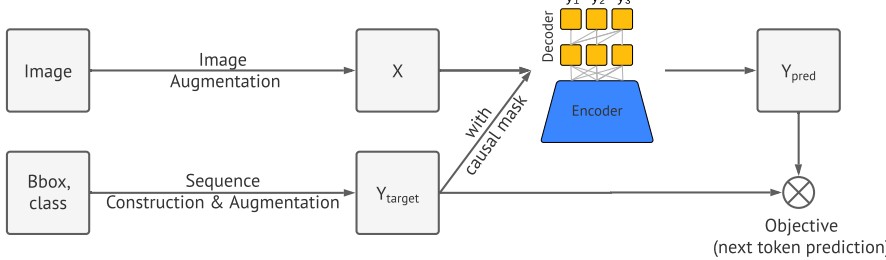

Figure 2: Major components of the Pix2Seq learning framework.

## 2.1 SEQUENCE CONSTRUCTION FROM OBJECT DESCRIPTIONS

In common object detection datasets, such as Pascal VOC (Everingham et al., 2010), COCO (Lin et al., 2014), and OpenImages (Kuznetsova et al., 2020), images have variable numbers of objects, represented as sets of bounding boxes and class labels. In Pix2Seq we express them as sequences of discrete tokens.

While class labels are naturally expressed as discrete tokens, bounding boxes are not. A bounding box is determined by two of its corner points (i.e., top-left and bottom-right), or by its center point plus height and width. We propose to discretize the continuous numbers used to specify the $x, y$ coordinates of corner points (similarly for height and width if the other box format is used). Specifically, an object is represented as a sequence of five discrete tokens, i.e. $[y_{\min}, x_{\min}, y_{\max}, x_{\max}, c]$, where each of the continuous corner coordinates is uniformly discretized into an integer between $[1, n_{\text{bins}}]$, and $c$ is the class index. We use a shared vocabulary for all tokens, so the vocabulary size is equal to number of bins + number of classes. This quantization scheme for the bounding boxes allows us to use a small vocabulary while achieving high precision. For example, a $600{\times}600$ image requires only 600 bins to achieve zero quantization error. This is much smaller than modern language models with vocabulary sizes of 32K or higher (Radford et al., 2018; Devlin et al., 2018). The effect of different levels of quantization on the placement of bounding boxes is illustrated in Figure 3.

With each object description expressed as a short discrete sequence, we next need to serialize multiple object descriptions to form a single sequence for a given image. Since order of objects does not matter for the detection task per se, we use a random ordering strategy (randomizing the order objects each time an image is shown). We also explore other deterministic ordering strategies, but we hypothesize that random ordering will work just as well as any deterministic ordering, given a capable neural net and autoregressive modeling (where the net can learn to model the distribution of remaining objects conditioned on those observed).

Finally, because different images often have different numbers of objects, the generated sequences will have different lengths. To indicate the end of a sequence, we therefore incorporate an EOS token. The sequence construction process with different ordering strategies is illustrated in Figure 4.

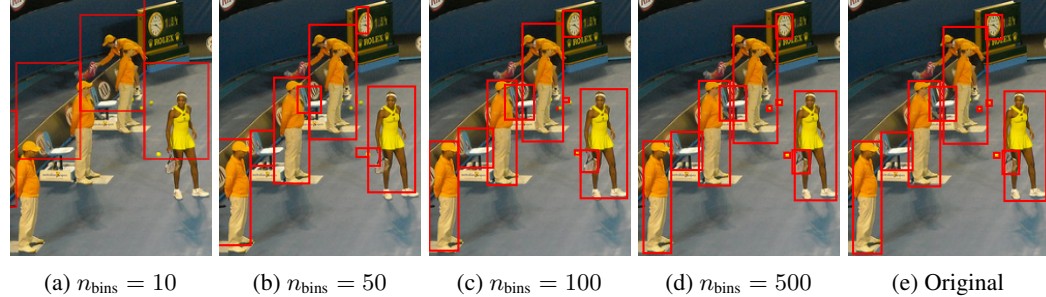

(a) $n_{\text{bins}} = 10$     (b) $n_{\text{bins}} = 50$     (c) $n_{\text{bins}} = 100$     (d) $n_{\text{bins}} = 500$     (e) Original

Figure 3: Applying the proposed discritization of bounding box on an image of $480 \times 640$. Only a quarter of the image is shown for better clarity. With a small number of bins, such as 500 bins (∼1 pixel/bin), it achieves high precision even for small objects.

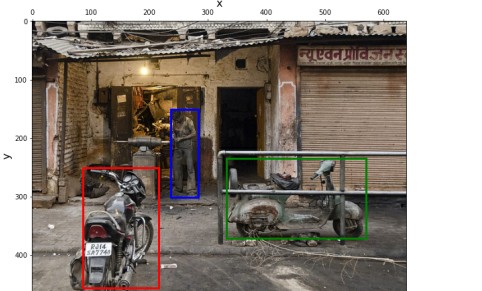

Random ordering (multiple samples):

327 370 653 444 1001   544 135 987 338 1004   508 518 805 892 1004   0
544 135 987 338 1004   327 370 653 444 1001   508 518 805 892 1004   0
508 518 805 892 1004   544 135 987 338 1004   327 370 653 444 1001   0

Area ordering:

544 135 987 338 1004   508 518 805 892 1004   327 370 653 444 1001   0

Dist2ori ordering:

544 135 987 338 1004   327 370 653 444 1001   508 518 805 892 1004   0

Figure 4: Examples of sequence construction with $n_{\text{bins}} = 1000$, and 0 is EOS token.

## 2.2 ARCHITECTURE, OBJECTIVE AND INFERENCE

Treating the sequences that we construct from object descriptions as a "dialect", we turn to generic architectures and objective functions that have been effective in language modeling.

**Architecture** We use an encoder-decoder architecture. The encoder can be a general image encoder that perceives pixels and encodes them into hidden representations, such as a ConvNet (LeCun et al., 1989; Krizhevsky et al., 2012; He et al., 2016), Transformer (Vaswani et al., 2017; Dosovitskiy et al., 2020), or their combination (Carion et al., 2020). For generation we use a Transformer decoder, widely used in modern language modeling (Radford et al., 2018; Raffel et al., 2019). It generates one token at a time, conditioned on the preceding tokens and the encoded image representation. This removes the complexity and customization in architectures of modern object detectors, e.g., bounding box proposal and regression, since tokens are generated from a single vocabulary with a softmax.

**Objective** Similar to language modeling, Pix2Seq is trained to predict tokens, given an image and preceding tokens, with a maximum likelihood loss, i.e.,

$$\text{maximize} \sum_{j=1}^{L} \boldsymbol{w}_j \log P(\tilde{\boldsymbol{y}}_j | \boldsymbol{x}, \boldsymbol{y}_{1:j-1}) \,, \tag{1}$$

where $\boldsymbol{x}$ is a given image, $\boldsymbol{y}$ and $\tilde{\boldsymbol{y}}$ are input and target sequences associated with $\boldsymbol{x}$, and $L$ is the target sequence length. $\boldsymbol{y}$ and $\tilde{\boldsymbol{y}}$ are identical in the standard language modeling setup, but they can also be different (as in our later augmented sequence construction). Also, $\boldsymbol{w}_j$ is a pre-assigned weight for $j$-th token in the sequence. We set $\boldsymbol{w}_j = 1, \forall j$, however it would be possible to weight tokens by their types (e.g., coordinate vs class tokens), or by the size of the corresponding object.

**Inference** At inference time, we sample tokens from model likelihood, i.e., $P(\boldsymbol{y}_j | \boldsymbol{x}, \boldsymbol{y}_{1:j-1})$. This can be done by either taking the token with the largest likelihood ($\arg\max$ sampling), or using other stochastic sampling techniques. We find that using nucleus sampling (Holtzman et al., 2019) leads to higher recall than $\arg\max$ sampling (Appendix C). The sequence ends when the EOS token is generated. Once the sequence is generated, it is straight-forward to extract and de-quantize the object descriptions (i.e., obtaining the predicted bounding boxes and class labels).

## 2.3 SEQUENCE AUGMENTATION TO INTEGRATE TASK PRIORS

The EOS token allows the model to decide when to terminate generation, but in practice we find that the model tends to finish without predicting all objects. This is likely due to 1) annotation noise (e.g., where annotators did not identify all the objects), and 2) uncertainty in recognizing or localizing some objects. While this only affects the overall performance by a small percentage (e.g., 1-2% in average precision), it has a larger effect on recall. To encourage higher recall rates, one trick is to delay the sampling of the EOS token by artificially decreasing its likelihood. However, this often leads to noisy and duplicated predictions. In part, this difficult trade-off between precision and recall is a consequence of our model being task agnostic, unaware of the detection task per se.

To mitigate the problem we simply introduce a sequence augmentation technique, thereby incorporating prior knowledge about the task. The target sequence $\tilde{\boldsymbol{y}}$ in conventional autoregressive language modeling (i.e., with no sequence augmentation) is the same as the input sequence $\boldsymbol{y}$. And all tokens in a sequence are real (e.g., converted from human annotations). With sequence augmentation, we instead augment input sequences during training to include both real and synthetic noise tokens. We also modify target sequences so that the model can learn to identify the noise tokens rather than mimic them. This improves the robustness of the model against noisy and duplicated predictions (particularly when the EOS token is delayed to increase recall). The modifications introduced by sequence augmentation are illustrated in Figure 5, and detailed below.

**Altered sequence construction** We first create *synthetic noise objects* to augment input sequences in the following two ways: 1) adding noise to existing ground-truth objects (e.g., random scaling or shifting their bounding boxes), and 2) generating completely random boxes (with randomly associated class labels). It is worth noting that some of these noise objects may be identical to, or overlapping with, some of the ground-truth objects, simulating noisy and duplicated predictions, as demonstrated

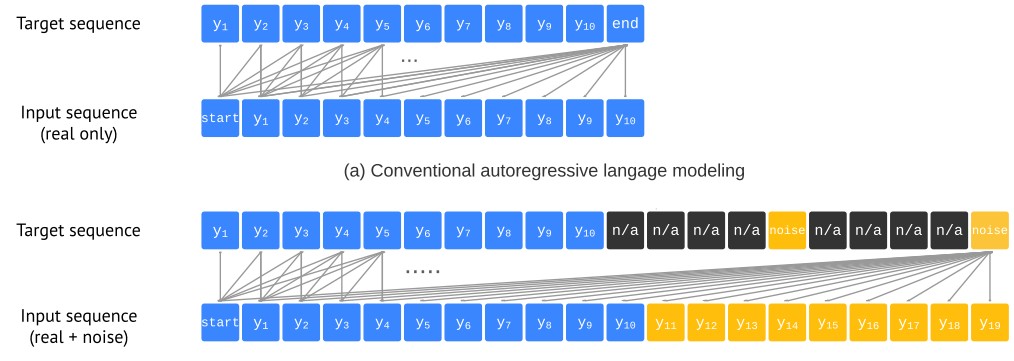

Figure 5: Illustration of language modeling with / without sequence augmentation. With sequence augmentation, input tokens are constructed to include both real objects (blue) and synthetic noise objects (orange). For the noise objects, the model is trained to identify them as the "noise" class, and we set the loss weight of "n/a" tokens (corresponding to coordinates of noise objects) to zero since we do not want the model to mimic them.

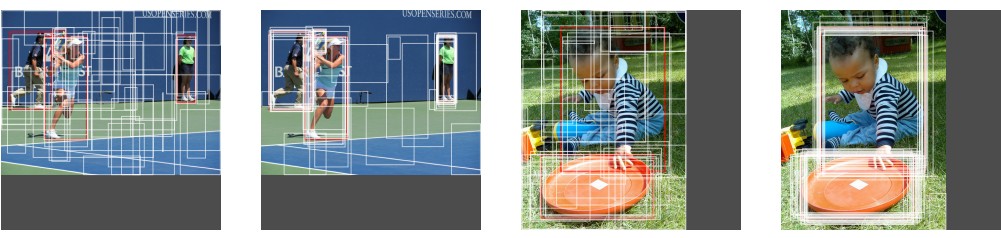

Figure 6: Illustrations of randomly sampled noise objects (in white), vs. ground-truth objects (in red).

in Figure 6. After noise objects are synthesised and discretized, we then append them in the end of the original input sequence. As for the target sequence, we set the target tokens of noise objects to "noise" class (not belonging to any of the ground-truth class labels), and the coordinate tokens of noise objects to "n/a", whose loss weights are set to zero, i.e., setting $\boldsymbol{w}_j = \mathbb{1}_{[\tilde{\boldsymbol{y}}_j \neq \text{"n/a"}]}$ in Eq 1.

**Altered inference** With sequence augmentation, we are able to substantially delay the EOS token, improving recall without increasing the frequency of noisy and duplicated predictions. Thus, we let the model predict to a maximum length, yielding a fixed-sized list of objects. When we extract the list of bounding boxes and class labels from the generated sequences, we replace the "noise" class label with a real class label that has the highest likelihood among all real class labels. We use the likelihood of the selected class token as a (ranking) score for the object.

## 3 EXPERIMENTS

### 3.1 EXPERIMENTAL SETUP

We evaluate the proposed method on the MS-COCO 2017 detection dataset (Lin et al., 2014), containing 118k training images and 5k validation images. To compare with DETR and Faster R-CNN, we report average precision (AP), an integral metric over multiple thresholds, on validation set at the last training epoch. We employ two training strategies: 1) *training from scratch* on COCO in order to compare fairly with the baselines, and also 2) *pretraining+finetuning*, i.e., pretrain the Pix2Seq model on a larger object detection dataset, namely Objects365 (Shao et al., 2019), and then finetune the model on COCO. Since our approach incorporates zero inductive bias / prior knowledge of the object detection task, we expect the second training strategy to be superior.

Table 1: Comparison of average precision, over multiple thresholds and object sizes, on COCO validation set. Each section compares different methods of the similar ResNet "backbone". Our models achieve competitive results to both Faster R-CNN and DETR baselines.

| Method | Backbone | #params | AP | $AP_{50}$ | $AP_{75}$ | $AP_S$ | $AP_M$ | $AP_L$ |
|---|---|---|---|---|---|---|---|---|
| Faster R-CNN | R50-FPN | 42M | 40.2 | 61.0 | 43.8 | 24.2 | 43.5 | 52.0 |
| Faster R-CNN+ | R50-FPN | 42M | 42.0 | 62.1 | 45.5 | 26.6 | 45.4 | 53.4 |
| DETR | R50 | 41M | 42.0 | 62.4 | 44.2 | 20.5 | 45.8 | 61.1 |
| Pix2seq (Ours) | R50 | 37M | **43.0** | 61.0 | 45.6 | 25.1 | 46.9 | 59.4 |
| Faster R-CNN | R101-FPN | 60M | 42.0 | 62.5 | 45.9 | 25.2 | 45.6 | 54.6 |
| Faster R-CNN+ | R101-FPN | 60M | 44.0 | 63.9 | 47.8 | 27.2 | 48.1 | 56.0 |
| DETR | R101 | 60M | 43.5 | 63.8 | 46.4 | 21.9 | 48.0 | 61.8 |
| Pix2seq (Ours) | R101 | 56M | **44.5** | 62.8 | 47.5 | 26.0 | 48.2 | 60.3 |
| Faster R-CNN | R50-DC5 | 166M | 39.0 | 60.5 | 42.3 | 21.4 | 43.5 | 52.5 |
| Faster R-CNN+ | R50-DC5 | 166M | 41.1 | 61.4 | 44.3 | 22.9 | 45.9 | 55.0 |
| DETR | R50-DC5 | 41M | **43.3** | 63.1 | 45.9 | 22.5 | 47.3 | 61.1 |
| Pix2seq (Ours) | R50-DC5 | 38M | **43.2** | 61.0 | 46.1 | 26.6 | 47.0 | 58.6 |
| DETR | R101-DC5 | 60M | **44.9** | 64.7 | 47.7 | 23.7 | 49.5 | 62.3 |
| Pix2seq (Ours) | R101-DC5 | 57M | **45.0** | 63.2 | 48.6 | 28.2 | 48.9 | 60.4 |

For training from scratch, we follow (Carion et al., 2020) using a ResNet backbone (He et al., 2016), followed by 6 layers of transformer encoder and 6 layers of (causal) transformer decoder (Vaswani et al., 2017). We resize images (with a fixed aspect ratio) so the longer side is 1333 pixels. For sequence construction, we use 2000 quantization bins, and we randomize the order of objects every time an image is shown. We append noise objects to real objects such that each image contains 100 objects in total, and hence a sequence length of 500. The model is trained for 300 epochs with a batch size of 128.

For pretraining on Objects365 dataset, we use similar settings as above with a few differences. Notably, instead of using the large 1333×1333 image size, we use a smaller image size of 640×640, and pretrain the models for 400K steps with batch size of 256. It is worth noting that this pretraining process is even faster than training from scratch due to the use of smaller image size. During the fine-tuning on COCO dataset, only a small number of epochs (e.g., 20 to 60 epochs) are needed to achieve good results. And we could use larger image size during fine-tuning as well. Due to the use of larger pretraining dataset, we also experiment with larger models with Vision Transformers (Dosovitskiy et al., 2020).

More details for both training strategies can be found in Appendix B. As for ablations, we use a ResNet-101 backbone with a smaller image size (the longer side is 640), and we train the model from scratch for 200 epochs.

## 3.2 MAIN COMPARISONS

**Training from scratch on COCO** We mainly compare with two widely recognized baselines: DETR and Faster R-CNN. DETR and our model have comparable architectures, but our Transformer decoder does not require learned "object queries" or separated heads for box regression and classification, since our model generates different types of tokens (e.g., coordinate and class tokens) with a single softmax. Faster R-CNN is a well established method, with optimized architectures such as feature-pyramid networks (FPN) (Lin et al., 2017a). Faster R-CNN is typically trained in fewer epochs than DETR or our model, likely because it explicitly incorporates prior knowledge of the task in the architecture itself. Thus we also include an improved Faster R-CNN baseline, denoted as Faster R-CNN+, from (Carion et al., 2020), where Faster R-CNN models are trained with the GIoU loss (Rezatofighi et al., 2019), train-time random crop augmentations, and the long `9x` training schedule.

Results are shown in Table 1, where each section compares different methods of the same ResNet "backbone". Overall, Pix2Seq achieves competitive results to both baselines. Our model performs comparably to Faster R-CNN on small and medium objects, but better on larger objects. Compared

Table 2: Average precision of finetuned Pix2seq models on COCO with different backbone architectures and image sizes. All models are pretrained on Objects365 dataset. As a comparison, our best model without pretraining obtains 45.0 AP (in Table 1) with image size of 1333×1333. The pretraining is with 640×640 image size while fine-tuning (a few epochs) can use larger image sizes.

| Backbone | # params | Image size during finetuning | | |
|---|---|---|---|---|
| | | 640×640 | 1024×1024 | 1333×1333 |
| R50 | 37M | 39.1 | 41.7 | 42.6 |
| R50-C4 | 85M | 44.7 | 46.9 | 47.3 |
| ViT-B | 115M | 44.2 | 46.5 | 47.1 |
| ViT-L | 341M | 47.6 | 49.0 | 50.0 |

with DETR, our model performs comparably or slightly worse on large and medium objects, but substantially better (4-5 AP) on small objects.

**Pretrain on Objects365 and finetune on COCO** As shown in Table 2, the performances of Objects365 pretrained Pix2Seq models are strong across various model sizes and image sizes. The best performance (with 1333 image size) is 50 AP which is 5% higher than the best model trained from scratch, and the performance holds up very well even with 640 image size. Notably, with a smaller image size used for pretraining, the pretrain+finetune process is faster than training from scratch, and also generalizes better. Both factors are crucial for training larger and better models.

### 3.3 Ablation on sequence construction

Figure 7a explores the effect of coordinate quantization on performance. For this ablation we consider images the longest size of which is 640 pixels. The plot indicates that quantization to 500 bins or more is sufficient; with 500 bins there are approximately 1.3 pixels per bin, which does not introduce significant approximation error. Indeed, as long as one has as many bins as the number of pixels (along the longest side of the image) there should be no significant error due to quantization of the bounding box coordinates.

We also consider different object ordering strategies in sequence construction during training. These include 1) random, 2) area (i.e., descending object size), 3) dist2ori (i.e., the distance of top-left corner of the bounding box to the origin), 4) class (name), 5) class + area (i.e., the objects are first ordered by their class, and if there are multiple objects of the same class, they are ordered by area), and 6) class + dist2ori. Figure 7b shows average precision (AP) and Figure 7c shows average recall (AR) at the top-100 predictions. Both in terms of precision and recall, the random ordering yields the best performance. We conjecture that with deterministic ordering, it may be difficult for the model to recover from mistakes of missing objects made earlier on, while with random ordering it would still be possible to retrieve them later.

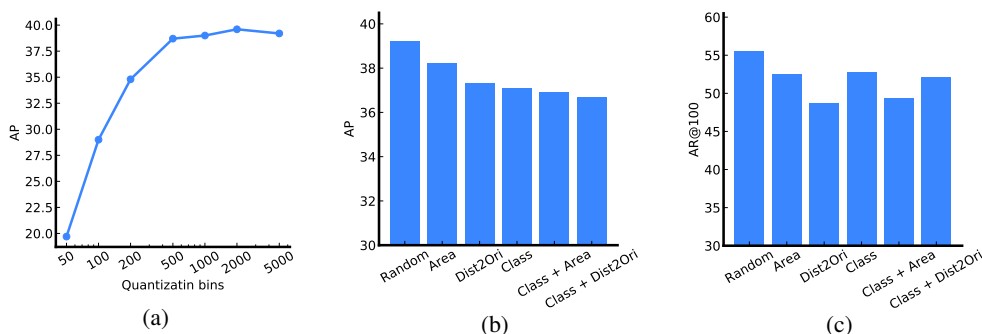

Figure 7: Ablations on sequence construction. (a) Quantization bins vs. performance. (b) and (c) show AP and AR@100 for different object ordering strategies.

### 3.4 ABLATION ON SEQUENCE AUGMENTATION

Here we study the impact of sequence augmentation (i.e., adding the noise objects) for both model training strategies: 1) training from scratch on COCO, and 2) pretraining on Objects365 and finetuning on COCO. Results for training from scratch w/wo sequence augmentation are shown in Figure 8, and we find that without sequence augmentation, the AP is marginally worse if one delays the sampling of EOS token during the inference (via likelihood offsetting), but the recall is significantly worse for the optimal AP. Table 3 shows similar results for pretraining+finetuning setting (where we set a loss weight of 0.1 on ending token instead of tuning their likelihood offset), and we find that AP is not significantly affected while recall is significantly worse without sequence augmentation. It is also worth noting that sequence augmentation is mainly effective during the fine-tuning.

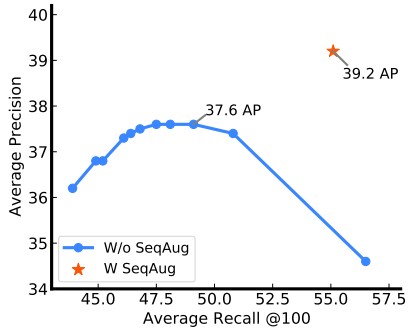

Figure 8: Impact of sequence augmentation on when training from scratch on COCO.

| SeqAug in Pretrain | SeqAug in Finetune | AP | AR@100 |
|:---:|:---:|:---:|:---:|
| ✗ | ✗ | 43.7 | 55.4 |
| ✗ | ✓ | 44.5 | 61.6 |
| ✓ | ✓ | 44.7 | 61.7 |

Table 3: Impact of sequence augmentation when pretraining on Objects365 and finetuning on COCO. Sequence augmentation has a major impact on average recall (@100) but a smaller influence on AP. Most improvements can be achieved during fine-tuning.

### 3.5 VISUALIZATION OF DECODER'S CROSS ATTENTION MAP

When generating a new token, the transformer decoder uses self attention over the preceding tokens and cross attention over the encoded visual feature map. Here we visualize the cross attention (averaged over layers and heads) as the model predicts a new token. Figure 9 shows cross attention maps as the first few tokens are generated. One can see that the attention is very diverse when predicting the first coordinate token (i.e $y_{min}$), but then quickly concentrates and fixates on the object.

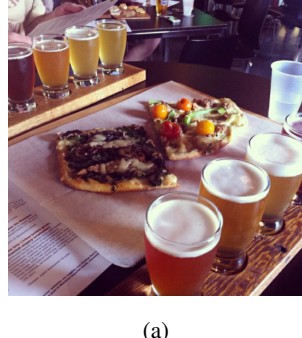 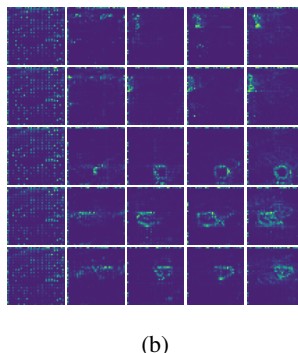 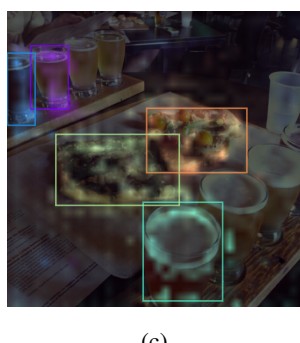

(a)      (b)      (c)

Figure 9: Decoder's cross attention to visual feature map when predicting the first 5 objects. (b) we reshape a prediction sequence of 25 into a 5x5 grid, so each row represents a prediction for 5 tokens $[y_{min}, x_{min}, y_{max}, x_{max}, c]$. The attention is diverse when selecting the first token of the object, then quickly concentrates on the object. (c) Overlay of the cross attention (when predicting the class token) on the original image.

## 4 RELATED WORK

**Object detection**. Existing object detection algorithms incorporate explicit prior knowledge about the task in their choice of architecture and loss function. To predict a set of bounding boxes, architectures of modern detectors are specifically designed to produce a large set of proposals (Girshick, 2015; Ren et al., 2015; Cai & Vasconcelos, 2018), anchors (Lin et al., 2017b), or window centers (Tian et al., 2019; Zhou et al., 2019). Non-maximum suppression (Bodla et al., 2017) is often required to prevent duplicate predictions. While DETR (Carion et al., 2020) avoids sophisticated bounding box proposals and non-maximum suppression, it still requires a set of learned "object queries", specially for object binding. These detectors all require sub-networks (or extra layers) separately for regressing bounding boxes and class labels. Pix2Seq avoids such complexities by having a generic image encoder and sequence decoder, with a single softmax for producing coordinate tokens and class labels.

Beyond architectures, the loss functions of existing detectors are also highly tailored for matching bounding boxes. For example, the loss function is often based on bounding box regression (Szegedy et al., 2013; Lin et al., 2017b), intersection over union (Rezatofighi et al., 2019), and set-based matching (Erhan et al., 2014; Liu et al., 2016; Redmon et al., 2016; Stewart et al., 2016; Carion et al., 2020). Pix2Seq avoids specialized losses, showing that a straightforward maximum likelihood objective with softmax cross entropy can work well.

Our work is also related to recurrent models in object detection (Stewart et al., 2016; Park & Berg, 2015; Romera-Paredes & Torr, 2016; Salvador et al., 2017; Ren & Zemel, 2017), in which the system learns to predict one object at a time. As above, both architecture and loss functions in these approaches are often tailored to the detection task. Furthermore, these approaches are not based on Transformers, and have not been evaluated against modern baselines on larger datasets.

**Language modeling**. Our work is inspired by recent success of modern language modeling (Radford et al., 2019; Raffel et al., 2019; Brown et al., 2020). Although originally intended for natural languages, the underlying methodology has been shown capable of modeling various sequential data, such as machine translation (Sutskever et al., 2014; Bahdanau et al., 2014), image captioning (Vinyals et al., 2015b; Karpathy & Fei-Fei, 2015; Xu et al., 2015), and many others (Vinyals et al., 2015a; Huang et al., 2018; Ramesh et al., 2021; Chen et al., 2021). Our work enriches this portfolio and shows that it works for even non-sequential data (by turning a set of objects into a sequence of tokens). We augment both input and target sequences for our model to incorporate task-specific prior knowledge; similar sequence corruption scheme have been used in language models (Devlin et al., 2018; Clark et al., 2020), and bear some similarity to noise-contrastive learning (Gutmann & Hyvärinen, 2010) and the discriminator in GANs (Goodfellow et al., 2014).

## 5 CONCLUSION AND FUTURE WORK

This paper introduces Pix2Seq, a simple yet generic framework for object detection. By casting object detection as a language modeling task, our approach largely simplifies the detection pipeline, removing most of the specialization in modern detection algorithms. We believe that our framework not only works for object detection, but can also be applied to other vision tasks where the output can be represented by a relatively concise sequence of discrete tokens (e.g., keypoint detection, image captioning, visual question answering). To this end, we hope to extend Pix2Seq as a generic and unified interface for solving a large variety of vision tasks.

A major limitation of our approach is that autoregressive modeling is expensive for long sequences (mainly during model inference). Practical measures to mitigate the issue includes: 1) stop inference when the ending token is produced (e.g., in COCO dataset, there are, in average, 7 objects per image, leading to a relatively small number of $\sim$35 tokens), 2) applying it to offline inference, or online scenarios where the objects of interest are relatively sparse (e.g. locate a specific object with language description). However, future work is needed to make it faster for real-time object detection applications. Another limitation is that the current approach for training Pix2Seq is entirely based on human annotation, and by reducing such dependence, it can enable the model to benefit from more unlabeled data.

## ACKNOWLEDGEMENTS

We specially thank Xiuye Gu for preparing the Objects365 dataset. We thank Mohammad Norouzi, Simon Kornblith, Tsung-Yi Lin, Allan Jabri, and Kevin Swersky for the helpful discussions.

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

## A  QUANTIZATION AND DEQUANTIZATION OF COORDINATES

Algorithm 1 and 2 illustrate the quantization and dequantization process of (normalized) coordinates.

| **Algorithm 1** Quantization of (normalized) coordinates | **Algorithm 2** Dequantization of discrete tokens of coordinates |
|---|---|
| ```python
def quantize(x, bins=1000):
  # x is a real number between [0, 1]
  # returns an integer between [0, bins-1]
  return int(x * (bins - 1))
``` | ```python
def dequantize(x, bins=1000):
  # x is an integer between [0, bins-1]
  # returns a real number between [0, 1]
  return float(x) / (bins - 1)
``` |

## B  TRAINING DETAILS

**Training from scratch on COCO**    For baseline architectures, we follow (Carion et al., 2020) using a ResNet backbone (He et al., 2016), followed by 6 layers of transformer encoder and 6 layers of (causal) transformer decoder (Vaswani et al., 2017). The main dimension of transformer is set to 256 with 8 attention heads, and the dimension of the feed-forward network is set to 1024. We use the stochastic depth (Huang et al., 2016) with a rate of 10% to reduce overfitting. Per (Carion et al., 2020), we also experiment with the DC5 variant of ResNet (Li et al., 2017), which increases the resolution of its output feature map by a factor of two.[2]

For image augmentation during training, we perform scale jittering with random crops (Ghiasi et al., 2021; Wu et al., 2019) with strength of $[0.1, 3]$. We resize images (with a fixed aspect ratio) so the longer side is 1333 pixels. Following (Howard, 2013; Chen et al., 2020a;b), we also use color distortion with a strength of 0.5. For sequence construction, we use 2000 quantization bins, and we randomize the order of objects every time an image is shown. We append noise objects to real objects such that each image contains 100 objects in total, and hence a sequence length of 500.

We train the entire network from scratch for 300 epochs with a batch size of 128. For each image in a mini-batch, we perform two independent augmentations, similar to (Hoffer et al., 2020), resulting in a 256 effective batch size, which we find helpful to reduce overfitting. We use AdamW optimizer (Kingma & Ba, 2014; Loshchilov & Hutter, 2018) with a learning rate of 0.003 and weight decay of 0.05. We use a learning rate warmup for 10 epochs and then linearly decay the learning rate over the course of training.

**Pretraining on Objects365**    We explore a wider range of architecture variants including both hybrid ResNet and transformer models (Carion et al., 2020), as well as pure transformers based on image patches (Dosovitskiy et al., 2020). The details of the architecture can be found in our released code. Since Objects365 dataset is much larger than COCO (1.7M images vs 118K images), we use a weaker image augmentation (scale jittering range of $[0.3, 2]$ for ViT backbones, and $[0.9, 1.2]$ for ResNet backbones) without color distortion. For sequence construction, we use 1000 quantization bins. And we still apply sequence augmentation with sampled noise objects added by default.

We use a smaller image size of $640{\times}640$, and pretrain the models for 400K steps with batch size of 256. We do not perform two augmentations per batch as in training from scratch. And we use a smaller learning rate of 0.001 with the same weight decay of 0.05. We use a cosine learning rate decay with a initial warmup of 20K steps.

As for the finetuning on COCO dataset, we use a batch size of 128 for ResNet backbones, and 64 for ViT backbones. Most models are finetuned for 60 epochs with a learning rate of $3e^{-5}$, but even fewer epochs yield similar results. We still use scale jittering with a range of $[0.3, 2]$ for image augmentation.

---

[2]Adding a dilation to the last ResNet stage and removing the stride from the first convolution of that stage.

## C ABLATION ON INFERENCE ($\arg\max$ VS NUCLEUS SAMPLING)

Nucleus sampling (Holtzman et al., 2019) has been applied to language modeling to reduce duplication and increase diversity in generated samples. Here we study its impact on sampling from our trained model.

Given the distribution $P(\boldsymbol{y}_j|\boldsymbol{x}, \boldsymbol{y}_{1:j-1})$, to apply nucleus sampling, we first define its top-$p$ vocabulary $V^{(p)} \subset V$ as the smallest set such that

$$\sum_{\boldsymbol{y}_j \in V^{(p)}} P(\boldsymbol{y}_j|\boldsymbol{x}, \boldsymbol{y}_{1:j-1}) \geq p. \tag{2}$$

Let $p' = \sum_{\boldsymbol{y}_j \in V^{(p)}} P(\boldsymbol{y}_j|\boldsymbol{x}, \boldsymbol{y}_{1:j-1})$, and we can re-calibrate the conditional likelihood as following for sampling the next token.

$$P'(\boldsymbol{y}_j|\boldsymbol{x}, \boldsymbol{y}_{1:j-1}) = \begin{cases} P(\boldsymbol{y}_j|\boldsymbol{x}, \boldsymbol{y}_{1:i-1})/p' & \text{if } \boldsymbol{y}_j \in V^{(p)} \\ 0 & \text{otherwise.} \end{cases} \tag{3}$$

We vary the hyper-parameter $p$ of nucleus sampling used in generating the output sequence (during inference). When $p = 0$, it corresponds to $\arg\max$ sampling, otherwise it samples from a truncated ranked list of tokens that has a cumsum larger or equal to $p$. In Figure 10, we see that use of nucleus sampling (with $p > 0$) improves object recall and thus also leads to better average precision. There is a relatively flat region of AP between 0.2 and 0.5, and we select $p$ to be 0.4 as our default value for other experiments.

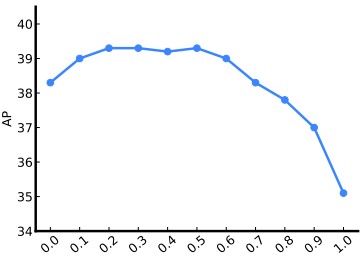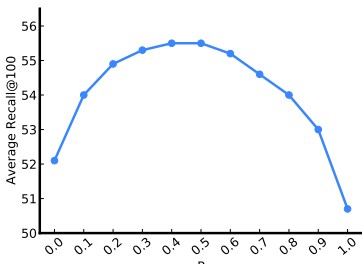

Figure 10: Varying parameter $p$ in nucleus sampling during inference results in different AP and AR. With $p = 0$, it is equivalent to argmax sampling. Sampling with $p > 0$ is helpful for increasing recall (and precision).

## D VISUALIZATION OF SIMILARITY AMONG COORDINATE TOKENS

In our model, bounding box coordinates are not represented as floating points, but encoded as discrete tokens. Here we study the similarity among these coordinate tokens via their embeddings. Note that the discrete coordinate tokens and class name tokens are in the same vocabulary and share the same embedding matrix. Specifically, we first slice the learned embedding matrix corresponding to coordinate tokens, and then compute the cosine similarity of embedding vectors for these coordinate tokens.

Figure 11 shows cosine similarity among embeddings of coordinate tokens. We can see that nearby coordinates have higher similarities in their token embeddings than far away ones. This emergent property of our model is likely due to the noises / uncertainties in bounding box annotations (i.e. a bounding box annotation is a random sample from a distribution over potential bounding boxes which encodes locality of coordinates).

## E THE ABILITY TO DIRECT THE ATTENTION WITH GIVEN COORDINATES

We explore the model's ability to *pay attention to a pointed region* specified via coordinates. We divide an image evenly into an $N \times N$ grid of rectangular regions, each specified by a sequence of

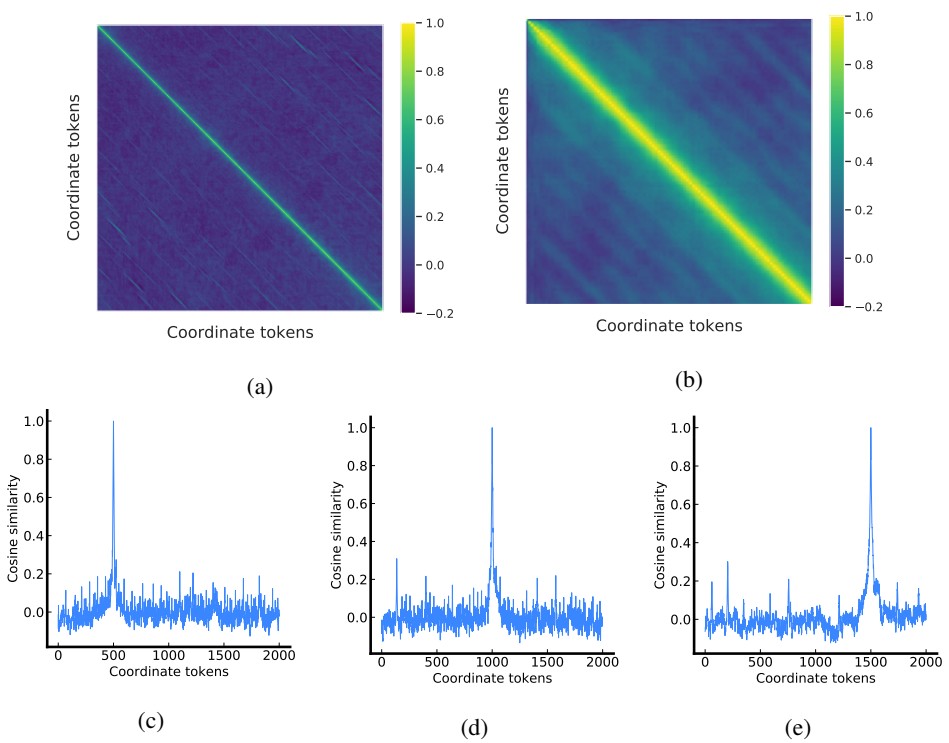

Figure 11: (a) Cosine similarity among embeddings of coordinate tokens. (b) is part of (a) covering only the first 100 tokens. (c), (d) and (e) are the 500-th, 1000-th and 1500-th rows of (a), respectively. Nearby coordinates have higher similarities in their token embeddings.

coordinates for its bounding box. We then visualize the decoder's cross attention to visual feature map after reading the sequence of coordinates for each region, i.e., $[y_{\min}, x_{\min}, y_{\max}, x_{\max}]$. We shuffle the pixels in the image to remove distraction from existing objects, and remove 2% of the top attentions for clarity. Interestingly, as shown in Figure 12, it seems the model can pay attention to the specified region at different scales.

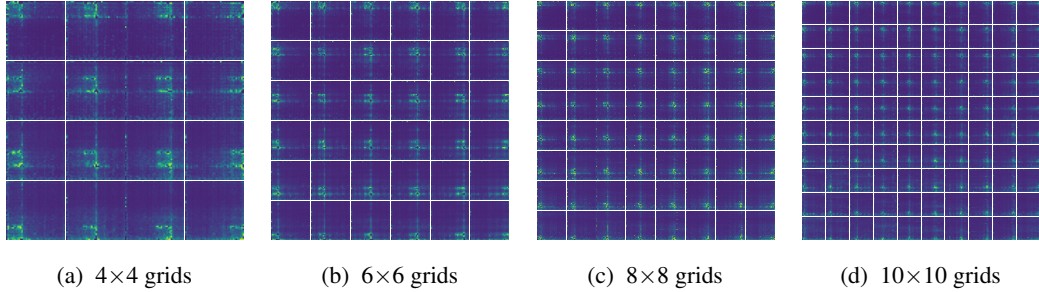

(a) 4×4 grids     (b) 6×6 grids     (c) 8×8 grids     (d) 10×10 grids

Figure 12: Each grid is a visualization of decoder's attention after reading a small sequence of coordinates, i.e., $[y_{\min}, x_{\min}, y_{\max}, x_{\max}]$. Visualization is done for grids of different sizes. The network learns to pay attention to pointed region at different scales.

## F  MORE VISUALIZATION ON DECODER'S CROSS ATTENTION

In Figure 13, we overlay the cross attention (when predicting the class token) on the original image for several other images, and it shows that the decoder pays the most attention to the object when predicting the class token.

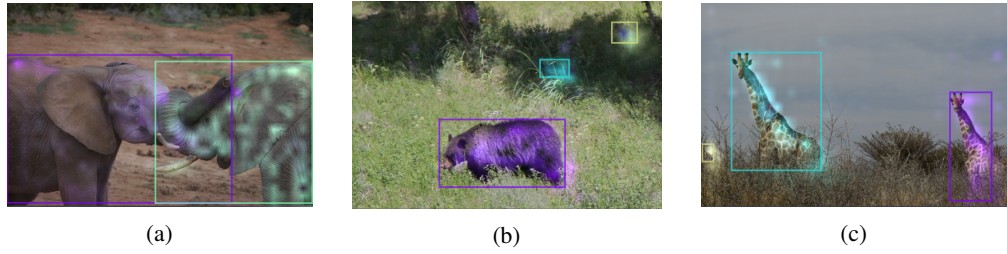

(a)                         (b)                         (c)

Figure 13: Visualization of Transformer decoder's cross attention (when predicting class tokens) conditioned on the given bounding boxes.

## G  VISUALIZATION OF DETECTION RESULTS

In Figure 14, we visualize detection results of one of Pix2seq model (with 46 AP) on a subset of images from COCO validation set that contain a crowded set of objects.

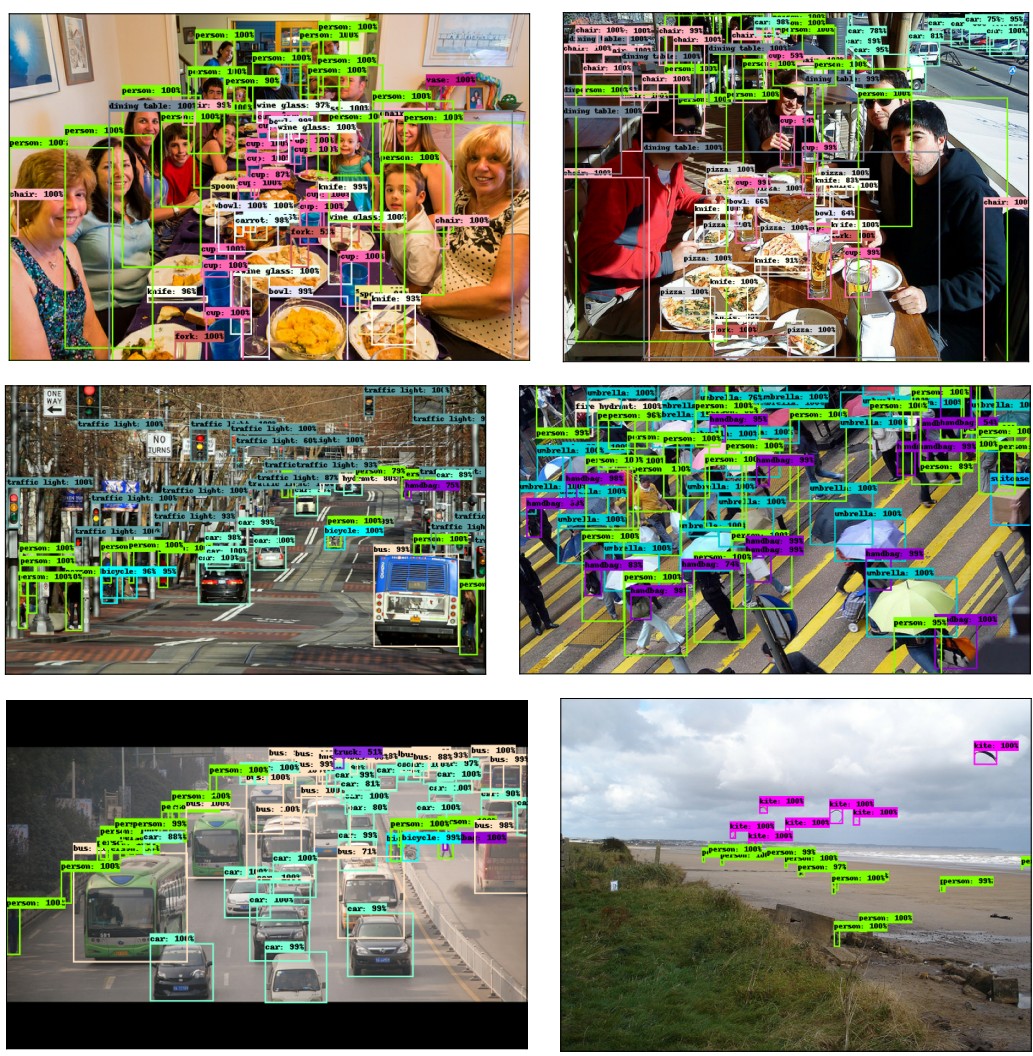

Figure 14: Examples of the model's predictions (at the score threshold of 0.5). Original images accessed by clicking the images in supported PDF readers.

