# OpenReview forum: "Pix2seq: A Language Modeling Framework for Object Detection"
_ICLR.cc/2022/Conference — ICLR 2022 Poster_

### Official Review · Reviewer_JMKG · 2021-10-30

**Correctness:** 4
**Technical Novelty And Significance:** 4
**Empirical Novelty And Significance:** 2
**Recommendation:** 8
**Confidence:** 5

**Main Review:**

**[Strength]**
1. The paper is generally well written and easy to understand.
2. The proposed method, Pix2Seq, firstly adopts a language model in object detection.
3. The proposed method is simple but achieves comparable results with existing methods.
4. There are various ablation studies for helping understand the embedding of language models on object detection.


**[Weakness]**
1. Inference Time

The main concern is inference time. Since the model has sequence prediction (“generate one token at a time”), it can take more time compared to existing models. Hence, adding inference time on Table 1 is needed.

2. Training Speed

Another main concern is training time, 300 epochs. This is one of the major concerns in DETR and may inherit the proposed method. It can be relaxed by using different architecture such as deformable DETR (Zu et al. 2021) or other efficient DETRs.

Zhu, Xizhou, et al. "Deformable detr: Deformable transformers for end-to-end object detection." ICLR. 2021.


3. Sequence Augmentation

3-1. The authors tackle the exploitation of prior knowledge of object detection in other methods. However, as mentioned in the second paragraph of section 2.3, the proposed method also exploits prior knowledge as a sequence augmentation and it is critical for the performance as shown in Figure 8.

3-2. On the other hand, a curve in Figure 8 is generated by “allowing the model to make more predictions.” However, there is no description of how to do that such as ignoring EOS until K steps. The authors should present or specify the exact performance for a model without Sequence Augmentation and not allow the mode to make more predictions.

4. EOS

Since the model uses a language model, it is possible to emit EOS between some coordinates or class tokens (e.g., y_min, x_min, y_max, EOS). How does the model deal with this? Does the model ignore the bounding box and stop the inference?


5. Attention Map

What does it mean by columns and rows in Figure 9 (b)? It seems that row means a different set of coordinates and classes (i.e., bbox) and the column means y_min, x_min, y_max, x_max, class. Although it is implicitly described in the context, I would recommend explicitly mentioning it.

6. Similarity among Coordinate Tokens

The authors only present the correlation matrix and say nearby coordinates have higher similarities in their token embedding. More explanation is needed. Also, the outputs have five (except EOS) different meanings (y_min, x_min, y_max, x_max, class). How is the correlation matrix generated?

7. Change Figure 14

 Currently, the authors add hyperlinks in each ‘url’. However, the authors also consider people who read the paper in a hard copy or offline. presenting the original image in the paper or even deleting the captions will be better.



**[Minor]**
1. DETR (Carion et al., 2020) does not have a box regression sub-network although it utilizes  GIoU loss. Please change the description in the first sentence of page 9, “These detectors”.

2. Change 43.2, 44.9 in Table 1 to plain text.


**[Recommendation]**
1. Move ablation study for image augmentation to Supplementary material.
=> Image augmentation is widely used in object detection as the author mentioned and there is no need to incorporate it in the main paper.

2. The author uses only 200 epochs for ablation study while the full model is 300 epochs, which is acceptable but still comparing at the same epochs (300) will be better.

3. In Section 3.3, “class (name)” seems “class + random.” For clarity, adding random or some description will be better.

4. Cross Attention Maps are really good. I suggest present instance or panoptic segmentation performance based on the cross attention map similar to DETR (Carion et al., 2020) paper.


**Summary Of The Paper:**


1. The paper tackles object detection by using the encoder-decoder structure of the language model, Pix2Seq.
2. The authors argue that the proposed method leverages prior knowledge for object detection less than existing object detection algorithms that exploit box regression, intersection-over-union, and so on.
3. Pix2Seq is based on maximum likelihood loss and employs sequence augmentation by adding dummy object bounding boxes to delay EOS (End Of Sequence)
Consequently, The proposed method achieves comparable results with existing methods.


**Summary Of The Review:**

In general, the proposed method is novel and gets good results. I believe it will get a huge attention and lead to a big change in computer vision community. However, there are a few concerns as mentioned above, especially inference time. I hope the authors resolve my concerns.

---

> ### Author Response · Authors · 2021-11-22
> **Response to Reviewer JMKG**
>
> We thank the reviewer for their constructive comments and suggestions. We address each below.
>
> **Inference/train cost**
>
> In the table below, we present inference flops break down for both our model and DETR model, which are based on similar architecture: ResNet-50 + Transformer encoder + Transformer decoder (DETR uses parallel decoder with 100 queries; Pix2seq uses autoregressive decoder with max of 500 tokens).
>
> | Image size | Compontents | GFLOPs (Pix2Seq) | GFLOPs (DETR) |
> |:--------------|:--------------------|:-----:|:---- :|
> | 640  (bsz100) | Resnet              | 33.4  | 33.4  |
> |               | Transformer encoder | 2.4   | 2.4   |
> |               | Transformer decoder | 0.32  | 1.1   |
> | 1333 (bsz20)  | Resnet              | 146.5 | 146.5 |
> |               | Transformer encoder | 18.3  | 18.3  |
> |               | Transformer decoder | 1.4   | 2.6   |
>
> We notice that most flops are dominated by the ResNet backbone, especially for large images. Although our decoder has smaller flops, the decoder (where the main architectural difference is) only accounts for a small fraction of the total flops.
>
> As for the actual inference time, it depends on the exact implementation, optimization and hardware.  We briefly tested a naive implementation of autoregressive decoding for image size of 1333, and it takes 294ms for 500 tokens while parallel decoder (100 queries) only takes 0.55ms (on a V100 GPU), despite the flops being similar. However, optimization can be applied to improve autoregressive decoding, such as the multi-query trick (https://arxiv.org/abs/1911.02150), quantization, distillation, better batching/caching, hardware aware optimization, etc. Additionally, our model can allocate compute dynamically according to the number of object instances in an image, so the actual sequence generated can be much shorter. For example, images in COCO dataset have around 7 object instances on average, so we may expect \~14X shorter sequences (\~35 tokens) on average compared to a fixed prediction size (500 tokens), given that we allow the model to decide when to end the prediction.
>
> As for training, the decoder is fully parallelizable with causal masks to enforce auto-regressive constraints, so the training cost of an autoregressive decoder and parallel decoder is almost identical.  We train for 300 epochs while DETR trains for 500 epochs, so overall our training time is less or comparable with DETR.
>
> **EOS**
>
> We use simple post-processing that splits the predicted sequence using a fixed format (4 coordinates and 1 class token) to get bounding boxes. So if the model makes an error by predicting EOS in the middle, it will lead to a false prediction. Typically we do not see such a phenomenon after the model is trained.
>
>
> **Attention Map**
>
> For Figure 9(b), we reshape a prediction sequence of 25 into a 5x5 grid for compact visualization, so each row represents a prediction for 5 tokens (y_min, x_min, y_max, x_max). We will clarify it in revision.
>
>
> **Similarity among Coordinate Tokens**
>
> The correlation is computed by taking the cosine similarity of embedding vectors for coordinate tokens. These coordinate tokens and class tokens are in the same vocabulary (consisting of integer numbers of coordinate bins and class indexes) and embedding matrix. Thus we slice the learned embedding matrix corresponding to coordinate tokens for computing the correlation matrix. The nearby coordinates share similar embeddings which is an emergent property, and we believe this is a result of uncertainty / noise in how annotations are realized (e.g. a bounding box may be randomly off a small number of pixels when they’re drawn). We will add more discussion.
>
>
> **Others**
>
> *Re “The authors should present or specify the exact performance for a model without Sequence Augmentation and not allow the mode to make more predictions”*: We will highlight the default model on Figure 8 in our revision.
>
> *Re “Change Figure 14”*: We will rephrase the “url” wording and put a caption to clarify that readers who want to look at the original image need to do so on a computer with the Internet.
>
> *Re “box regression sub-network”*: DETR has linear layers for classes and box regression (Listing 1 of the paper), which technically counts as a sub-network so we could clarify it.

---

> > ### Comment · Reviewer_JMKG · 2021-11-22
> > **Two concerns from author's response**
> >
> > Thank the authors for dealing with the comments. I still have a few concerns as I am writing as fast as possible so that the authors can respond.
> >
> > First of all, GFLOPS does not represent the actual computation time. It only counts addition and multiplication, which means that there is no difference between sequential and parallel computations. Considering the actual inference time, according to the author’s response, the sequential model (Pix2Seq) is 500 times slower. Even if compared with 28 FPS (from the original DETR paper), it is 8 times slower. Although the author mentioned several tricks to reduce the speed, most of them are also applicable to parallel decoders as well, and some gaps still remain.
> >
> > Secondly, regarding the usage of prior knowledge, would it be possible to present the results without sequential augmentation compared with DETR and other methods? I understand there are only 17 hours left but it would be helpful to convince people that the model less exploits prior knowledge. Since the authors argue Pix2Seq is task-agnostic (without prior knowledge), it is critical.
> >
> > Hope that the authors have enough time to respond to my concerns.

---

> > > ### Author Response · Authors · 2021-11-22
> > > **Response to Reviewer JMKG**
> > >
> > > We thank the reviewer for a quick follow-up.
> > >
> > > *Re actual inference time*: We agree some of the mentioned optimizations can be applied to parallel decoder as well, but they may not be as helpful compared to autoregressive decoder (e.g. the multi-query trick reduces repeated sequential computation and significantly speeds autoregressive decoder, but it may not help as significantly for a parallel decoder since the computation is parallelizable anyway). There are also some optimization that can be used in our setting (e.g. dynamic computation / sequence length) *cannot* be applied to parallel decoder as the latter requires a fixed number of queries. Overall, we expect the gap between autoregressive and parallel decoder to significantly shrink with those optimizations, or to a level that may be negligible given the encoder may dominate the computation. But as this may require extra amount of work to realize, we will explicitly discuss this aspect in the revision.
> > >
> > > *Re model without sequence augmentation*: We would like to first clarify that what we argue as task-agnostic in Pix2seq are the sequence interface, architecture and objective function, as they are not specialized to object detection task. The data preprocessing / augmentation (image or sequence) component *is* task-specific, and will remain this way as long as different tasks represent data in different ways (e.g., coordinates, labels, texts, audio, pixels). Per reviewer's question on our model performance without sequence augmentation, it can be found in the ablation experiments in Figure 8(b), which we will highlight in revision more: without sequence augmentation, it gets 37.6 AP at best, while with sequence augmentation, it reaches 39.2 AP.

---

> > > > ### Comment · Reviewer_JMKG · 2021-11-26
> > > > **Response to Authors**
> > > >
> > > > Thank the authors for the comprehensive response.
> > > >
> > > > **[Actual Inference Time]**
> > > > 1. The manuscript cannot be changed right now but I will believe the author will keep their promise that revises the manuscript adding actual inference time and improved inference speed as they mentioned.
> > > >
> > > > **[Augmentation]**
> > > > 1. If the 'task-agnostic' contribution is limited in the architecture and object function as the authors mentioned above response (and the last paragraph of introduction), they should revise the abstract "Beyond the use of task-specific data augmentations, our approach makes minimal assumptions about the task" or, "framework can achieve competitive results on COCO" in the introduction since the performance partly lean on task-specific augmentation. 37.6 is **not** competitive performance compared with Faster- R-CNN.
> > > > 2. Moreover, I still highly recommend adding results without sequential augmentation in Table 1 or mentioning the model uses sequential augmentation in the main results again since people who skim through the paper might think that Table 1 results are acquired from purely task-agnostic methods.
> > > >
> > > > **[Others]**
> > > > 1. After reading other reviewers' comments and responses from the authors, I also agree that the model is hard to practically use and the motivation for using language model for object detection is unclear although the authors have argued that this is "showcase a standardized interface for which vision tasks can be addressed".
> > > > 2. On the other hand, it is less clear that how Pix2Seq can be used in and have an impact on the computer vision community even though the authors mentioned that it can be applied to other vision tasks that produce low-bandwidth output (Conclusion) considering below:
> > > > 1) the architecture resembles DETR as Reviewer Wvff mentioned.
> > > > 1-1) Pix2Seq makes reconsideration the existence of box regression similar to DETR (NMS). However, without augmentation, the performance gap is too huge.
> > > > 2) the inference time should be inefficient compared with parallel architecture.
> > > > => Since the paper is a kind of proof of concept, I recommend the authors describe the potentials of language model in computer vision more precisely and practically.
> > > >
> > > > In general, I still have a positive viewpoint as the authors firstly present language models in object recognition although it seems less motivated and has less clear potentials.

---

> > > > > ### Author Response · Authors · 2021-11-26
> > > > > **a quick clarification on number, and the difference in injecting prior knowledge in data augmentation vs in modeling**
> > > > >
> > > > > Just to clarify, 37.6 AP (without seq augmentation) is obtained using smaller image resolution and shorter training schedule, so is 39.2 AP (with seq augmentation), for keeping our ablation settings consistent. For this reason, this 37.6 AP is not directly comparable with other numbers (e.g. Faster R-CNN) in Table 1. It's also worth to note that seq augmentation is part of data preprocessing, similar to image augmentation, as they are done before modeling. Image augmentation is widely used in most computer vision systems, so we do not think seq augmentation should be treated much differently (just for it is something new here). We believe there is quite a difference between injecting prior knowledge in data augmentation vs in modeling (architecture and objective), pertaining data/model/task scaling, generalization across tasks, and so on. But since this line of research (on reducing task prior in modeling for computer vision) is still in its infancy, its potential cannot be fully demonstrated in a single paper of ours, so we appreciate the reviewer's sincere comments and being forward looking.

---

### Official Review · Reviewer_bYa9 · 2021-11-01

**Correctness:** 3
**Technical Novelty And Significance:** 2
**Empirical Novelty And Significance:** 3
**Recommendation:** 6
**Confidence:** 3

**Main Review:**

The submission has merit and a potential to receive considerable attention by the research community due to having an unconventional (but not necessary novel ) view point to the object detection problem with good results. However, I have few major comments on the paper presentation, novelty, limitation, experiments and ablation study listed below:


1-	Presentation: I am not really convinced about the story of the paper and the way it is presented and motivated. As I reflected in the abstract, in my view (though I might misunderstood the technical details), this framework simply formulates the object detection problem as 1) a sequence prediction problem using 2) a transformer-based autoregressive model and 3) the key idea is to ensure each outputted bounding box can be formulated as a set of the token (i.e. by discretising the bounding box states to a set of bins). To this end, the link to a language modelling is weak, strange and arbitrary for the presentation. Considering my viewpoint to the proposed approach (the notes 1-3), the strong argument against limitation of the existing detection techniques and their formulation for the motivation of this work also seem to be invalid as this framework does not really address them.
2-	Novelty: Formulating the object detection as the bounding box sequence prediction (note 1) is not very novel (e.g. Stewart et al 2016), and in comparison in this framework, the previous frameworks did not have advantage of using very powerful backbone and decoders as the proposed framework. Bounding box regression is also known to be harder task than classification and to this end, few recent methods have approximated the task by discretising the spaces (e.g. Qiu et al. Offset bin classification network for accurate object detection, CVPR 2020).
3-	Limitation: while this strategy may work on 2D object detection framework with axis align bounding boxes, it is not clear how it will perform on (and also computationally scales for) the detection problems such as non-axis aligned object detection (e.g. detection from Satellite image) or 3D object detections (with 6 DoF), where the number of discretising bins (token) can increase exponentially.
4-	Experiments: It is hard to compare which components help more to the Pix2seq framework’s good results from Table 1, e.g. (a) network model: a better decoder (a large transformers encoder and decoder) compared to Faster R-CNN (few MLP layers in the second stage),  (b) Formulation: autoregressive sequence prediction instead of tensor prediction in Faster R-CNN or set prediction in DETR (c.f. Rezatofighi et al. arxiv 2020) or (c) loss variation: avoiding regression loss by discretising the bounding box representation and using softmax loss similar to classification. It is also meaningful to include both inference variations of the framework in Table 1 (argmax sampling & nucleus sampling)

One  minor comment:

The Number of parameters in Table 1 should include both backbone and decoder. It would be great if FLOP is also reported

I can understand why random ordering might perform better than a handcrafted (potentially inconsistent) deterministic ordering, but this random strategy also can be sub-optimal. The chain rule decomposition in Eq. 1 can be written in L! ways. While the weight in neural network should learn a joint representation agnostic to these output orders for the same input x, learning this number of combination may not be achievable by a random sampling. In the other sequential techniques, e.g. Vinyals et al, Order Matters, 2015  and Stewart at al. 2016, the best permutation is selected dynamically during training stage by solving an assignment problem between all the predictions and GT before the loss calculation.



**Summary Of The Paper:**

This paper presents a new framework for object detection by casting the problem as an (auto-encoder based) auto-regressive sequence prediction using a CNN based backbone as the encoder to encode visual features and transformer-based encoder & decoder (c.f. section 3.1) as the decoder to predict each bounding box sequentially. All the bounding boxes in an image are generated auto-regressively conditioned on the image features from the backbone and previous predictions (c.f. Eq. 1). The key idea in this paper is to output each axis-aligned bounding box as the set of tokens representing the possible bin locations for its two corners and then to use cross-entropy loss (SoftMax) along with the class token to predict each detection.  The best model is trained by the random ordering of bounding boxes. The approach achieved competitive results on the challenging COCO dataset, compared to Faster R-CNN and DETR.

**Summary Of The Review:**

The submission carries an important message for the object detection problem and the framework results are reasonably good, But the paper presentation should be substantially improved to be a reflective of the contribution.  Limitation of this formulation should be clearly elaborated and the ablation study should contain insightful results for the contribution of each proposed module (formulation, architecture or loss). To this end, I rate this submission borderline

---

> ### Author Response · Authors · 2021-11-22
> **Response to Reviewer bYa9**
>
> We thank the reviewer for their constructive comments. We address each below.
>
> **Presentation & novelty**
>
> Our goal is to showcase a standardized interface for which vision tasks can be addressed (through the lens of object detection). We chose the interface to be language-like, and we use the same language modeling architecture and objective to tackle it. As such, we respectfully disagree with the reviewer that “the link to a language modeling is weak”, as language modeling is key to our proposal. Using object detection, a core vision task, as a proof of concept, our message is that for any vision task that can be described effectively in terms of discrete tokens, eg as <image, seq>, it can potentially use the same architecture and objective, subject to per-task data preprocessing (including data augmentation, which may become less critical when trained with large-scale data).
>
> Although it may not have been obvious, the key to our proposal is not the autoregressive sequential modeling of object bounding boxes per se, but rather, that we can define a generic loss on an output expressed as a sequence. We believe that when some other generic (potentially non-autoregression) loss is proposed for sequence modeling (e.g. for language modeling), it is likely that one will be able to apply it to our framework as well (replacing the autoregressive loss). As such, the modeling does not necessarily need to be sequential, even if the output is formulated as a sequence.  Also, while there are existing methods (e.g. Stewart et al 2016) that formulate object detection as sequential prediction problem, they still require specific architecture for bounding boxes and class labels (while in pix2seq they are just tokens in a unified vocabulary), and they require a task-specific bounding box matching loss (while in pix2seq one can just use the same cross entropy loss for all tokens of bounding boxes and class labels).
>
> Unlike existing methods, ours is the first paper to our knowledge that shows that a generic language model can adequately solve object detection tasks without inductive bias (task prior) in both architecture and the objective function. While data augmentation is used to improve performance at present, this requirement may be lifted in the future by training on large-scale  data, or by other improved techniques.
>
>
> **Limitation**
>
> We do not think that “number of discretising bins (token) will increase exponentially” when extending this to 3D or other scenarios, as we just need to add extra tokens of descriptions. For example, we could use [y_min, x_min, z_min, y_max, x_max, z_max, …] to describe 3D bounding boxes, and having the exact same vocabulary size. The number of possible sequences grows exponentially while the number of actual individual tokens grows linearly (as we use tokens for the elements of the sequence rather than one token for an entire bounding box).
>
> **Experiments**
>
> We agree that the question raised by the reviewer, i.e., whether some proposals in our work can be borrowed by existing methods and how that changes their performances, is a good research question. However, we respectfully disagree that this is a fair criticism for the paper, whose goal is to provide the first proof of a concept for a new approach, one that represents a substantial departure from conventional architectures and loss functions that one usually sees in the object detection literature. We have provided a good number of ablations to our method (e.g., ordering, quantization, augmentation), to see what are crucial to our model design, but borrowing techniques for the baselines (e.g., applying autoregressive prediction for Faster R-CNN or DETR as suggested by the reviewer) in order to (systematically) figure out what makes our method work so well is non-trivial.  Although interesting, this is a large enough endeavor that we consider it outside of the scope of the current paper. It should be explored in future work.
>
> **Others**
>
> *Re “FLOPS”*: the FLOPs are comparable for both our model and corresponding DETR model, we will clarify it in the revision.
>
> *Re “Random order factorization”*: We believe with a strong model, it should be possible to model a set as a sequence with random order. The conclusion of Vinyals et al is reached using RNNs while we are using Transformers so there may not be conflict. Intuitively, a strong model should be adequate with random order, as it just needs to predict (uniformly) the remaining objects given those that are predicted.

---

> > ### Comment · Reviewer_bYa9 · 2021-11-25
> > **Not enough insight yet**
> >
> > Thanks to the author(s)  for their comprehensive response. While some of my comments are addressed, my major concerns still stay
> >
> > It is not yet very clear that in comparison with the existing detection frameworks, the good detection results are reflective of a better neural model (transformers vs CNN/or RNN (e.g. Stewart et al 2016)), formulation (Autoregressive/sequential model vs Tensor/Set prediction model) or better loss modelling (discrete bin loss for regression vs continuous regression loss, e.g. l1 or GIoU). The current ablations do not address this critical concern making the judgment about the superiority of the proposed method hard. It is not convincing to claim only over the entire model as proof of the concept and showcase its performance only in COCO dataset without proper insightful analysis.
> >
> > " it may not have been obvious, the key to our proposal is not the autoregressive sequential modelling of object bounding boxes per se" I believe the author(s) should ensure the clarity of the presentation so that the key contribution is obvious for the readers. the presentation is also the key for a paper to be accpeted

---

> > > ### Author Response · Authors · 2021-11-25
> > > **Addressing the remaining concerns of Reviewer bYa9**
> > >
> > > We appreciate the reviewer providing comments on the remaining concerns. We address them here:
> > >
> > > **Model variants and insights**
> > >
> > > For a new approach that significantly departs from the convention, there are always more questions than answers, but it doesn't mean the new approach (without exhaustively exploring all potential variants) lacks insights. As an example, the original Transformer model encapsulates many new & old components such as positional encoding, attention, MLP, layer norm and skip connection. However, explorations of potential variants of Transformers appear years after the original work, and in many papers not just in one. We believe our ablations are self-contained and provide enough insights for our proposal, exploring all potential variants of the system should be future work. Specifically, we have compared variants that reviewer mentioned above as much as possible in our paper, as follows.
> > >
> > > (1) "neural model": we follow DETR on architecture use (CNN+Transformer encoder + Transformer decoder) and also match configurations (e.g., layers, dimension), so it is *directly* comparable to DETR (our main baseline); while we have used multiple variants (ResNet 50 vs 101, standard vs DC5) as suggested by DETR, using other architectures would be an orthogonal issue and also makes comparisons to baselines unfair. (2) "formulation": compared to set prediction model in DETR, our autoregressive modeling with simple and generic cross entropy loss can work just as well (sometimes better), without requiring task-specific definition of loss function (GIOU loss based on bounding boxes is critical for DETR, accounts for ~5 AP difference in their ablation). (3) "loss modeling": we consider not needing GIOU and l1 loss as one major benefit of our model so we don't see why/how we should change our model to use GIOU/l1 loss as suggested by the reviewer; it's also non-trivial, if possible, to incorporate discretization or non-regression loss in DETR. We hope this addresses the reviewer's concern, but we are also open to concrete suggestions of ablations if the reviewer still has concerns here.
> > >
> > > **Key contribution and presentation**
> > >
> > > we believe we have done a decent job at conveying the key contribution in the paper (as summarized in the title, abstract and TL'DR: We demonstrated that object detection can be tackled by simply training a language model conditioned on pixel inputs, where the interface, architecture, and loss function we used are agnostic to object detection task, showing a potential for generalizing beyond a single task). It's inevitable that readers from different domains have (very) different takes, and some may think autoregressive modeling is the key contribution, while it is just one of the many important components of the proposed system. We are happy to further clarify this in the revision and/or other suggestions the reviewer may have on avoiding this confusion, but we believe our major message is clear.

---

> > > > ### Comment · Reviewer_bYa9 · 2021-11-29
> > > > **Suggested baselines**
> > > >
> > > > Thanks to the author(s) for the response. I hope my comments are considered as constructive suggestions for improving the paper quality.
> > > >
> > > > "concrete suggestions of ablations" Few suggestions for the potential baselines for comparison could be
> > > > 1- A baseline for formulation: Pix2seq, but with regression losses such as l1, l2 or/and GIoU. This experiment can provide insights into how the autoregressive/sequential model can perform in comparison with the set prediction model (very fair comparison with DETR if exactly the same regression loss is used and the model is almost similar)
> > > > 2- A baseline for the effect of neural model: Pix2seq, but using non-transformer-based autoregressive decoders (e.g. recurrent models). This experiment will show the contribution of the transformer model for making such detection formulation working
> > > >
> > > >  I know the submission cannot be further changed now. But I hope to see those experiments in the final version of the paper if accepted

---

### Official Review · Reviewer_Hjdy · 2021-11-02

**Correctness:** 4
**Technical Novelty And Significance:** 4
**Empirical Novelty And Significance:** 2
**Recommendation:** 8
**Confidence:** 5

**Main Review:**



STRENGTHS

S1 The study is novel and scientifically interesting in order to further understand the potential of the language model task beyond textual inputs and outputs.

S2 The loss function is the classic negative log likelihood, so it is not specifically designed for the task, as in other works from the state of the art.

S3 Proposes a approach to encode the continuous coordinates of the bounding boxes to discrete tokens. This simplifies the decoder architecture, compared to DETR.

S4 The work does not require the non-maximum supression post-processing, as in DETR.

S5 The proposed data augmentation approach to avoid the early EOS or repeated detections, observed in previous works, is novel.

S6 It includes ablation studies on the size of the quatization bins and sequence ordering, as well as insightful visualizations of the cross-attention maps.

S7 The manuscript is well written, with multiple figures that facilitate comprehension.



WEAKNESSES

W1 The obtained metrics in accuracy do not achieve state of the art, but are competitive.

W2 While the work motivates that existing approaches often focus on specific domains (self-driving cars, medical image analysis or agriculture), the results presented focus only on the COCO bechmark. Providing results for some of these specific domains would provide better insights about the potential of Pix2Seq in the referred myriad of domains.


W3a The weight w_j for the tokens in Equation 1 is defined but never used or tested. An experimentals analysis of its impact should be included to justify it.

W3b The loss function seems to compare input and target sequences, but actually the order in which the bounding boxes+class labels are generated should not be taken into account. That is, the loss should be invariant to the ordering of the predicted objects. It seems that it will penalize in the proposed set ups, while previous work have used the Hungarian algorithm to match predicted and ground truth detections before computing the loss. It is unclear why Pix2Seq is not adopting this same paradigm and whether it has a negative effect in the results.

W4 An ablation study or disucss about the impact of the sequence augmentation is needed, as this is a mmain contribution of this work.

W5 An analysis or discussion of the computational memory & requirements with respect to Faster R-CNN & DETR is needed to obtain a full picture beyond the accuracy results only.


MINOR COMMENTS

C1 In figure 9, what are the columns ? It seems to be the cross-attention maps when predicting the 4 coordinates + class. If so, then the whole output sequence of 25 tokens is the result of reading row by row ? More guidance to the reader may be helpful.

C2 Given the sucess of seq2seq models for image generation (eg, iGPT) when trained with large amounts of data, one woders whether training by just more data results would actually reach state of the art. For example, the OpenImages dataset already provides much more data could may be used to explore the gains in this direction.


**Summary Of The Paper:**

Pix2Seq provides a simple approach for object detection understood as a sequence generation task, in this case, of the coordinates of the bounding box and the object class. The architecture resembles DETR, but simplifies the decoder thanks to the formulation as a sequence decoding of discretized tokens. The manuscript proposes a scheme to discretize the bounding box coordinates in histogram bins and proposes data augmentation for the sequence, which  address two observed limitations when predictig sequences from images: early EOS and repetition of objects. Results on MS-COCO indicate slightly better results than DETR with the more simple architecture.

**Summary Of The Review:**

The proposed ideas are novel and, from my perspective, valuable enough for their publication despite not obtaining new state of the art results for the task. There is already a significant body of works that have addressed the analysis of images as a sequential process in which a predicted token coditions that posterior predictions. Pix2Wav simplifies DETR and improves its performance a bit. However, there are still some doubts listed in the weaknesses that should be clarified before making a final decision.

---

> ### Author Response · Authors · 2021-11-22
> **Response to Reviewer Hjdy**
>
> We thank the reviewer for their constructive comments. We address each below.
>
> **Datasets**
>
> As we focus on the object detection task in this work, we follow the convention of most object detection literatures, which is mostly based on COCO. We think COCO is a challenging dataset and also a well-benchmarked datasets so we can compare with baselines (e.g., DETR) easily. We agree it is a good idea to evaluate the approach over many detection datasets, but it is not trivial to do so for a fair comparison across different approaches, so we leave this as a future work.
>
> **Token weights**
>
> We do use the token weights as a way to define the loss. Mainly, the weights for coordinate tokens of noise objects (in target sequence) are set to zero so the model won’t mimic those noises. We haven’t done ablation on this since the consequence of setting all token weights to 1 after sequence augmentation seems obvious;  that is, the model will predict both good and noise objects, and this leads to strictly worse performance.
>
> **Loss and ordering**
>
> We use a sequence autoregressive loss rather than set loss as we formulate the output of the object detection task in a unified language interface, which has the benefit of being task-agnostic. A set loss (as in DETR and others) is seemingly more natural for an object detection task, but it requires the definition of matching and bounding box/class loss, and this typically needs task-specific knowledge to define, and cannot be directly generalized to other tasks.
>
> For autoregressive loss with random ordering, one can view this as marginalizing over permutations, so, in expectation, it should still learn the set, given a sufficiently powerful model like Transformers. Intuitively, the model should be able to do the task, as it just needs to predict the remaining objects given those that have already been  predicted. The model cannot achieve zero loss obviously due to random orders, but it should achieve minimal loss when it learns the correct distribution. Thus, we believe this is a generic and adequate objective.
>
>
> **Inference/train cost**
>
> In the table below, we present inference flops break down for both our model and DETR model, which are based on similar architecture: ResNet-50 + Transformer encoder + Transformer decoder (DETR uses parallel decoder with 100 queries; Pix2seq uses autoregressive decoder with max of 500 tokens).
>
> | Image size | Compontents | GFLOPs (Pix2Seq) | GFLOPs (DETR) |
> |:--------------|:--------------------|:-----:|:---- :|
> | 640  (bsz100) | Resnet              | 33.4  | 33.4  |
> |               | Transformer encoder | 2.4   | 2.4   |
> |               | Transformer decoder | 0.32  | 1.1   |
> | 1333 (bsz20)  | Resnet              | 146.5 | 146.5 |
> |               | Transformer encoder | 18.3  | 18.3  |
> |               | Transformer decoder | 1.4   | 2.6   |
>
> We notice that most flops are dominated by the ResNet backbone, especially for large images. Although our decoder has smaller flops, the decoder (where the main architectural difference is) only accounts for a small fraction of the total flops.
>
> As for the actual inference time, it depends on the exact implementation, optimization and hardware.  We briefly tested a naive implementation of autoregressive decoding for image size of 1333, and it takes 294ms for 500 tokens while parallel decoder (100 queries) only takes 0.55ms (on a V100 GPU), despite the flops being similar. However, optimization can be applied to improve autoregressive decoding, such as the multi-query trick (https://arxiv.org/abs/1911.02150), quantization, distillation, better batching/caching, hardware aware optimization, etc. Additionally, our model can allocate compute dynamically according to the number of object instances in an image, so the actual sequence generated can be much shorter. For example, images in COCO dataset have around 7 object instances on average, so we may expect \~14X shorter sequences (\~35 tokens) on average compared to a fixed prediction size (500 tokens), given that we allow the model to decide when to end the prediction.
>
> As for training, the decoder is fully parallelizable with causal masks to enforce auto-regressive constraints, so the training cost of an autoregressive decoder and parallel decoder is almost identical.  We train for 300 epochs while DETR trains for 500 epochs, so overall our training time is less or comparable with DETR.

---

> > ### Author Response · Authors · 2021-11-22
> > **remaining response to Reviewer Hjdy**
> >
> > **Others**
> >
> > *Re “An ablation study or discuss about the impact of the sequence augmentation is needed”*: We presented the ablation of sequence augmentation vs no sequence augmentation in Figure 8(b). We have not ablated the detailed hyper-parameters in sequence augmentation as they are quite detailed and potentially, they could be searched/tuned by techniques like AutoML (similar to AutoAugment), but we will consider picking some important ones to ablate in a revision of the paper as this may be useful for readers.
> >
> > *Re “In figure 9, what are the columns ? More guidance to the reader may be helpful”*:  We reshape a prediction sequence of 25 into a 5x5 grid for compact visualization, so each row represents a prediction for 5 tokens (y_min, x_min, y_max, x_max, class). We will clarify it in revision.
> >
> > *Re “whether training by just more data results would actually reach state of the art”*: Yes. Good point.  We also believe that larger scale pretraining should improve the model further. However, it may be non-trivial to scale it up (both computation-wise and approach-wise), so we definitely plan to explore this in the future.

---

> > > ### Comment · Reviewer_Hjdy · 2021-11-27
> > > **Score increased thaks to satisfactory and fair answers**
> > >
> > > Thanks to the authors by addressing the points in my review. I especially appreciate that the authors clearly indicate and quantify one of the weaknesses in terms of inference time, given the auto-regressive nature of the approach, compared to the parallel decoder of DETR. Still, the discussion of this issue and, the one about why the loss function may be appropriate despite not trying to match the ordering of the sequences, are valuable and enriching for scientific discussion. In encourage the authors to introduce these ideas in the final version.
> > >
> > > Overall, I increase my score given that most of my doubts are clarified.

---

### Official Review · Reviewer_Wvff · 2021-11-03

**Correctness:** 4
**Technical Novelty And Significance:** 3
**Empirical Novelty And Significance:** 2
**Recommendation:** 6
**Confidence:** 3

**Main Review:**

Pros:

1. This paper proposed a novel idea for object detection. Unlike most previous work, the proposed pix2seq model converts the coordinates and labels of objects into a sequence of token and leverage a seq2seq model to complete the prediction.

2. To mitigate the overfitting issue, the authors proposed a few techniques such as training sequence augmentation via noisy annotations. It turns out to be a helpful way to improve object detection performance.

3. The experimental results show that the proposed method achieve comparable performance to two strong baselines, including Faster R-CNN and DETR. Further ablation studies indicate that the sequence augmentation indeed helps and some visualizations align with the intuition behind the model.

Cons:

1. The proposed pix2seq is novel in that it uses a sequence generation model to predict object locations and classes, getting rid of the sophisticatedly designed architecture such as Faster R-CNN. However, it still resembles previous works like DETR in that they both exploited an encoder-decoder architecture. DETR decodes the predictions in parallel while the proposed model does it sequentially. Then what are the benefits and unique advantages of modeling it as a sequential decoding problem? I agree that it has the potential to unify different sequential decoding models. However, it is hard to tell in this paper and it is also an open question whether we want to unify all tasks into a sequence generation pipeline because localizing objects seems not to be a sequential task and does not heavily rely on temporal information (Fig. 7(a)(b) shows that random ordering during training obtains the highest mAP.)

2. According to the experimental results, the performance is comparable to Faster R-CNN and DETR while the training/inference may be much more time-consuming. Since the authors did not report the time cost of the proposed method during training and inference. My impression is that such a sequential model may introduce more time cost than parallel ones such as Faster R-CNN or DETR. This leads to the same question asked above. What are the main benefits of modeling object detection as a sequence generation task?

3. The authors claimed that the proposed method is unlike previous highly specified or heavily optimized ones. However, it seems that there is no free lunch. To avoid overfitting, the authors need to use sequence augmentation. To get better decoding results, the model also relies on a better sampling strategy, i.e., nucleus sampling. Even with these two techniques, I do not see a higher ceiling of performance starting from a low floor due to overfitting, which is unlike the one we observed in Vision Transformers for image recognition. I would like to hear more from the authors about what could be potentially applied to further improve the performance.

4. I am curious about whether the proposed method can generalize well across different image sizes during inference. Accordingly, the authors used normalized coordinates. I guess this will not be a big issue. But still, it would be great to see whether the proposed method can perform multi-scale inference giving different image sizes.

5. In the proposed method, the decoder predicts the quantized coordinate tokens given input feature map and preceding tokens. To precisely predict the locations, the model needs to map the heat map into discrete tokens. I am wondering whether the model heavily relies on positional embeddings. If yet, what kind of positional embedding is used for the encoder, and how this will affect the final performance?

**Summary Of The Paper:**

In this paper, the authors proposed a novel way of formulating object detection as a seq2seq task. Given an image, the model first uses a visual encoder the obtain a feature map and then sequentially decode the coordinates and label through a decoder. Different from previous conventional object detection pipelines, such as Faster R-CNN, the proposed method does not rely much on the prior knowledge or assumption about the task but lets the model learns by itself from training data. To avoid overfitting, the authors proposed a few techniques, including data augmentation and a more sophisticated decoding strategy. The experimental results show that it can achieve comparable performance with two established baselines, Faster R-CNN and DETR. These results indicate that even with a simple seq2seq pipeline, the model can be still on par with previous strong baseline methods.

**Summary Of The Review:**

I think this paper proposed a novel idea to reformulate object detection to a sequence generation problem. It provides us with a new perspective to think about conventional vision tasks. However, from the paper, I do see several drawbacks of the proposed method. Without seeing strong proofs, it is hard to determine whether the proposed method is a good one for generic object detection. As such, I have a general concern that this paper can bring us a new viewpoint but not new insight to solve object detection problems.

---

> ### Author Response · Authors · 2021-11-22
> **Response to Reviewer Wvff**
>
>
> We thank the reviewer for their constructive comments. We address each below.
>
> **Benefits of the formulation**
>
> Our goal is to showcase a standardized interface for which vision tasks can be addressed (through the lens of object detection). We chose the interface to be language-like, and we use the same language modeling architecture and objective to tackle it. Using object detection, a core vision task, as a proof of concept, our message is that for any vision task that can be described effectively in terms of discrete tokens, eg as <image, seq>, it can potentially use the same architecture and objective, subject to per-task data preprocessing (including data augmentation, which may become less critical when trained with large-scale data).
>
> This may not be obvious, but the key to our proposal is not the autoregressive sequential modeling of object bounding boxes per se, but rather, that we can define a generic loss on an output expressed as a sequence. We believe that when some other generic (potentially non-autoregression) loss is proposed for sequence modeling (e.g. for language modeling), it is likely that one will be able to apply it to our framework as well (replacing the autoregressive loss). As such, the modeling does not necessarily need to be sequential, even if the output is formulated as a sequence.
>
> The reviewer may also be concerned about the ordering used in our current autoregressive loss. One can view an autoregressive loss with random ordering in terms of marginalizing over  permutations, so that, in expectation, it should still learn the set, given a powerful enough model like Transformers. Intuitively, the model should be able to do the task as it just needs to predict the remaining objects given those already predicted. The model cannot achieve zero loss obviously due to random orders, but it should achieve minimal loss when it learns the correct distribution. Thus, we believe this is a generic and adequate objective.
>
>
> **Positional encoding**
>
> We use a learned positional encoding for its simplicity. Specially, the learnable positional embedding is a tensor of [height, width, channel] where height and width are determined by the output feature map of ResNet. The learnable positional embedding is summed with lower-dim projected ResNet output for Transformer processing. We believe other forms of positional encoding should work similarly, but we haven’t done detailed comparisons as we mainly follow DETR architecture to make it comparable (note that we didn’t use their exact positional encoding as it requires masking ResNet features and we wanted to do something simpler).
>
> **Inference/train cost**
>
> In the table below, we present inference flops break down for both our model and DETR model, which are based on similar architecture: ResNet-50 + Transformer encoder + Transformer decoder (DETR uses parallel decoder with 100 queries; Pix2seq uses autoregressive decoder with max of 500 tokens).
>
> | Image size | Compontents | GFLOPs (Pix2Seq) | GFLOPs (DETR) |
> |:--------------|:--------------------|:-----:|:---- :|
> | 640  (bsz100) | Resnet              | 33.4  | 33.4  |
> |               | Transformer encoder | 2.4   | 2.4   |
> |               | Transformer decoder | 0.32  | 1.1   |
> | 1333 (bsz20)  | Resnet              | 146.5 | 146.5 |
> |               | Transformer encoder | 18.3  | 18.3  |
> |               | Transformer decoder | 1.4   | 2.6   |
>
> We notice that most flops are dominated by the ResNet backbone, especially for large images. Although our decoder has smaller flops, the decoder (where the main architectural difference is) only accounts for a small fraction of the total flops.
>
> As for the actual inference time, it depends on the exact implementation, optimization and hardware.  We briefly tested a naive implementation of autoregressive decoding for image size of 1333, and it takes 294ms for 500 tokens while parallel decoder (100 queries) only takes 0.55ms (on a V100 GPU), despite the flops being similar. However, optimization can be applied to improve autoregressive decoding, such as the multi-query trick (https://arxiv.org/abs/1911.02150), quantization, distillation, better batching/caching, hardware aware optimization, etc. Additionally, our model can allocate compute dynamically according to the number of object instances in an image, so the actual sequence generated can be much shorter. For example, images in COCO dataset have around 7 object instances on average, so we may expect \~14X shorter sequences (\~35 tokens) on average compared to a fixed prediction size (500 tokens), given that we allow the model to decide when to end the prediction.
>
> As for training, the decoder is fully parallelizable with causal masks to enforce auto-regressive constraints, so the training cost of an autoregressive decoder and parallel decoder is almost identical.  We train for 300 epochs while DETR trains for 500 epochs, so overall our training time is less or comparable with DETR.

---

> > ### Author Response · Authors · 2021-11-22
> > **remaining response to Reviewer Wvff**
> >
> > **Others**
> >
> > *Re: what could be potentially applied to further improve the performance.* In this work we simply follow DETR for neural architecture design. However, with many recent advances in transformer-based architectures. We believe a better architecture would improve the performance. Also, the models trained in this paper are trained from scratch, but we believe that large-scale pre-training will also benefit the model’s performance.
> >
> > *Re: Multi-scale inference giving different image sizes.*  Our current architecture allows inference on different image sizes up to 1333 but they are padded so the maximum side is 1333. By adjusting the architecture such as positional encoding masking, one may be able to directly perform inference on smaller image sizes.

---

> > > ### Comment · Reviewer_Wvff · 2021-11-29
> > > **thanks for the feedbacks**
> > >
> > > Thanks to the authors for the thorough feedback. The authors answered all my questions and addressed my concerns to some extent.
> > >
> > > First of all, I buy the argument from the authors that the proposed pixel2seq model does not necessarily use a sequential decoding to obtain the sequence. As pointed out by the authors, the main merit of this work is to represent object detection predictions as a sequence of tokens. I suggest the authors to highlight this comment in the revision and also demonstrate this with some experiments. For example, what if the authors use DETR-like queries but output a sequence of tokens. rather than object class ids and box coordinates as in DETR.
> > >
> > > Second, the inference time reported by the authors indeed brings some concerns about the proposed method, at least in the current version. From the comparisons, I do not see much benefit of using a sequential decoding against a parallel one, though the authors claimed that it can be accelerated by many potential techniques. If the main contribution of this work is reformulating object detection as a prediction of token sequences, this drawback may probably dismiss if the authors can show a parallel decoding also works to some extent. Again, the authors should add the training/inference time cost in the revision.
> > >
> > > Third, for multi-scale inference, I am not very clear about the authors' answer. Did the authors try adjusting positional encoding masking to accommodate the different input sizes? Besides this adjustment, what else should be changed or modified to adapt to smaller input sizes? The authors should report some results for decreased input image sizes to show that this is trivial enough.
> > >
> > > thanks,

---

> > > > ### Author Response · Authors · 2021-11-30
> > > > **Clarification on multi-scale inference**
> > > >
> > > > We thank the reviewer for further feedback. Re *multi-scale inference* question - to deal with different image sizes at inference, we currently use a simple strategy, which is to rescale images so that the longer side is 1333. This is similar to, but simpler than, the strategy that baselines used (scaling shorter side to 800 and longer side less than 1333). Since baselines do not perform multi-scale inference evaluation, we also do not perform such evaluation. We think how to deal with smaller images efficiently at inference time is an orthogonal issue on the architecture, and it is not a specific issue for us as DETR also predicts (normalized) coordinates of the boxes despite using a regression and GIOU loss. But since the reviewer asked, we can provide a few thoughts on this - in order to support efficient multi-scale inference at (actual) smaller image sizes (unpadded/unscaled images), we need to either train on actual smaller images (can be difficult due to mini-batching), or use some masking mechanism so image sizes in training batches can be padded even. This makes it difficult to compare among baselines, and also makes models more complicated, so instead we opted for simplicity in this work, not supporting multi-scale inference.

---

### Official Review · Reviewer_kHNj · 2021-11-03

**Correctness:** 4
**Technical Novelty And Significance:** 4
**Empirical Novelty And Significance:** 4
**Recommendation:** 8
**Confidence:** 4

**Main Review:**

#### Overview

- The idea is novel, the proposed approach is simple and elegant and the paper is well written with most of the technical details with strong experimental results. I really appreciate the authors advocating a totally new approach for object detection based on the intuition that *``if a neural network knows about where and what the objects are, we just need to read them out''*. The proposed model is good proof of that intuition.

- The proposed model and training objective is more general compared to prior models on object detection. Compared to Faster-RCNN, pix2seq gets rids of bounding box proposals, ROI pooling which is highly customized for detection tasks. Compared to the more recent Detr, pix2seq gets rid of the object queries and set-based matching loss. This is a huge improvement towards a more unified model for vision tasks.

- Besides all those pros, there are a few missing details in the paper that needs further clarification (details in the weakness section.

#### Strength

- The idea of *providing a language interface to a wide range of vision tasks* is novel and the proposed model is simple, elegant, and achieves strong performance on object detection benchmark.

- Language modeling with sequence augmentation is novel and useful to encourage higher recall rates.

- Extensive and informative ablation studies on ResNet variant, #bins, different object ordering strategies, image scale augmentation, and sequence augmentation.

#### Weakness

- Position encoding (PE) should be very important for the proposed approach. However, the paper didn't discuss what PE is used in the paper (e.g. absolute, learned, relative) and how to use them (e.g. adding to seq embedding or adding to key, value in attention). The discussion and ablation study on different PEs will be super useful for the reader to replicate the model.

- In altered sequence construction, there are two ways to synthetic the noise sequence. I wonder what is the percentage of different synthetic noise sequences used in the paper?

- In figure 5, the noise token is start with <y_11> but not <end> token. This seems to break the auto-regressive sequence construction. I wonder is there any specific reason to do this?

- What is the inference time of the proposed model. It is known that the auto-regressive model is slow at the decoding stage, but comparing it to the Detr model will be informative for the readers.

- Instance or semantic segmentation (with variable sequence length) seems a natural extension to the proposed model. I wonder is there any comment on this task using the proposed approach?


**Summary Of The Paper:**

This paper proposed a language modeling framework (pixel2seq) for object detection. The authors cast object detection as language modeling tasks that use a sequence of tokens (x1, y1, x2, y2, c) to describe the bounding box and train an auto-regressive decoder to generate the target sequence. Compared to the existing approach (Faster-RCNN, Detr), the proposed model uses a more general architecture and loss function and achieves state-of-the-art performance on COCO datasets.

**Summary Of The Review:**

The idea is novel, the proposed approach is simple and elegant and the paper is well written with most of the technical details with strong experimental results.

---

> ### Author Response · Authors · 2021-11-22
> **Response to Reviewer kHNj**
>
> We thank the reviewer for their constructive comments. We address each, in turn, below.
>
> **Positional encoding**
>
> We use a learned positional encoding for its simplicity. Specially, the learnable positional embedding is a tensor of [height, width, channel] where height and width are determined by the output feature map of ResNet. The learnable positional embedding is summed with lower-dim projected ResNet output for Transformer processing. We believe other forms of positional encoding should work similarly well, but we haven’t done detailed comparisons as we mainly follow DETR architecture to make it comparable (note that we didn’t use their exact positional encoding as it requires masking ResNet features and we wanted to do something simpler).
>
> **Synthetic noise**
>
> To determine the number of each type of noise we uniformly sample an integer K from [0, max noise objects], and then sample K duplicated noises. The rest of the noise objects will be random noise. We plan to release the code in the near future; in this way the  implementation details will become much clearer.
>
> For Figure 5(b), if we put an <end> token before the noise tokens, the model would learn to predict, and only predict, noise tokens after seeing the <end> token. In our proposed construction, the model cannot easily take a shortcut to detect the noises. During training, our input/target sequence setup differs from a conventional autoregressive setting, but since the model only conditions on inputs generated up to a given point, the decoding/inference is not affected, and remains autoregressive.
>
> **Inference/train cost**
>
> In the table below, we present inference flops break down for both our model and DETR model, which are based on similar architecture: ResNet-50 + Transformer encoder + Transformer decoder (DETR uses parallel decoder with 100 queries; Pix2seq uses autoregressive decoder with max of 500 tokens).
>
> | Image size | Compontents | GFLOPs (Pix2Seq) | GFLOPs (DETR) |
> |:--------------|:--------------------|:-----:|:---- :|
> | 640  (bsz100) | Resnet              | 33.4  | 33.4  |
> |               | Transformer encoder | 2.4   | 2.4   |
> |               | Transformer decoder | 0.32  | 1.1   |
> | 1333 (bsz20)  | Resnet              | 146.5 | 146.5 |
> |               | Transformer encoder | 18.3  | 18.3  |
> |               | Transformer decoder | 1.4   | 2.6   |
>
> We notice that most flops are dominated by the ResNet backbone, especially for large images. Although our decoder has smaller flops, the decoder (where the main architectural difference is) only accounts for a small fraction of the total flops.
>
> As for the actual inference time, it depends on the exact implementation, optimization and hardware.  We briefly tested a naive implementation of autoregressive decoding for image size of 1333, and it takes 294ms for 500 tokens while parallel decoder (100 queries) only takes 0.55ms (on a V100 GPU), despite the flops being similar. However, optimization can be applied to improve autoregressive decoding, such as the multi-query trick (https://arxiv.org/abs/1911.02150), quantization, distillation, better batching/caching, hardware aware optimization, etc. Additionally, our model can allocate compute dynamically according to the number of object instances in an image, so the actual sequence generated can be much shorter. For example, images in COCO dataset have around 7 object instances on average, so we may expect \~14X shorter sequences (\~35 tokens) on average compared to a fixed prediction size (500 tokens), given that we allow the model to decide when to end the prediction.
>
> As for training, the decoder is fully parallelizable with causal masks to enforce auto-regressive constraints, so the training cost of an autoregressive decoder and parallel decoder is almost identical.  We train for 300 epochs while DETR trains for 500 epochs, so overall our training time is less or comparable with DETR.
>
>
> **Others**
>
> *Re: Instance or semantic segmentation.* We selected object detection as a proof of concept task in this work as it is a canonical computer vision task. Nevertheless, we think this framework can be extended to other tasks like instance segmentation, or keypoint prediction, by formulating tasks as <image, seq> pairs. There are probably many ways of achieving this, so we leave a detailed discussion of this issue to future exploration.

---

### Decision · Program_Chairs · 2022-01-20

**Decision:**

Accept (Poster)

**Comment:**

This paper proposes an elegant approach to object detection where an encoder network reads in an image and a decoder network outputs coordinate and category information via a sequence of textual tokens. This method does away with several object detection specific details and tricks such as region proposals and ROI pooling. The paper received positive reviews from all reviewers who agreed that this formulation of object detection was novel and provided a new perspective that may transfer to other computer vision tasks. One common concern amongst reviewers was the slow inference time due to the sequential nature of the decoder -- and this concern was a central point of discussion between the authors and reviewers. My takeaway from this discussion is that this model is certainly slower than traditional computer vision models that can generate boxes in parallel. The slowdown however, is image dependent. Less cluttered environments require shorter output sequences. Moreover, such a model can easily be applied to concept localization, e.g. "Locate the horses", in which cases one can expect fewer objects of the desired category, and hence acceptable inference speeds. Importantly, the contributions of this paper are noteworthy in spite of the proposed architecture having the drawback of being slow. Given this, I recommend accepting this paper for its merits.